# Two-step membrane binding by the bacterial SRP receptor enable efficient and accurate Co-translational protein targeting

Yu-Hsien Hwang Fu[1], William Y C Huang[2], Kuang Shen[1†], Jay T Groves[2], Thomas Miller[1], Shu-ou Shan[1*]

[1]Division of Chemistry and Chemical Engineering, California Institute of Technology, Pasadena, United States; [2]Department of Chemistry, University of California at Berkeley, Berkeley, United States

**Abstract** The signal recognition particle (SRP) delivers ~30% of the proteome to the eukaryotic endoplasmic reticulum, or the bacterial plasma membrane. The precise mechanism by which the bacterial SRP receptor, FtsY, interacts with and is regulated at the target membrane remain unclear. Here, quantitative analysis of FtsY-lipid interactions at single-molecule resolution revealed a two-step mechanism in which FtsY initially contacts membrane via a Dynamic mode, followed by an SRP-induced conformational transition to a Stable mode that activates FtsY for downstream steps. Importantly, mutational analyses revealed extensive auto-inhibitory mechanisms that prevent free FtsY from engaging membrane in the Stable mode; an engineered FtsY pre-organized into the Stable mode led to indiscriminate targeting in vitro and disrupted FtsY function in vivo. Our results show that the two-step lipid-binding mechanism uncouples the membrane association of FtsY from its conformational activation, thus optimizing the balance between the efficiency and fidelity of co-translational protein targeting.

*For correspondence: sshan@ caltech.edu

Present address: [†]Department of Biology, Whitehead Institute for Biomedical Research and Massachusetts Institute of Technology, Cambridge, United States

Competing interests: The authors declare that no competing interests exist.

## Introduction

Co-translational targeting of nascent membrane and secretory proteins by the Signal Recognition Particle (SRP) is an essential and universally conserved pathway that mediates the proper localization of almost 30% of the proteins encoded by the genome. The universally conserved core of SRP contains an SRP54 protein (termed Ffh in bacteria) tightly bound to an SRP RNA. A methionine-rich M-domain in SRP54 (or Ffh) recognizes the signal sequences on nascent polypeptides emerging from translating ribosomes. The GTPase domain, termed NG-domain, in SRP54 (or Ffh) interacts with a highly homologous NG-domain in SR (termed FtsY in bacteria), thus delivering the ribosome-nascent chain complex (RNC) to the eukaryotic endoplasmic reticulum (ER) or the bacterial plasma membrane (*Walter and Johnson, 1994*; *Cross et al., 2009*). At the membrane, the RNC is unloaded from the SRP•SR complex onto the Sec61p (or SecYEG) translocon, via which the nascent protein is either integrated into or translocated across the membrane.

Extensive work on the bacterial SRP showed that the delivery of RNCs to the target membrane is actively regulated by the substrate and translocon complex (*Akopian et al., 2013b*; *Zhang and Shan, 2014*). During each targeting cycle, the SRP•FtsY complex sequentially transitions between three conformational states, *early*, *closed* and *activated*, that culminate in their reciprocal GTPase activation (*Zhang et al., 2008*, *Zhang et al., 2009*). The *early* complex is a GTP-independent labile intermediate that stably forms only when SRP is loaded with RNCs bearing SRP-dependent substrate

proteins (*Zhang et al., 2010*; *von Loeffelholz et al., 2013*; *Bradshaw et al., 2009*). This stabilization enables the *early* intermediate to preferentially rearrange to the stable *closed* complex rather than dissociating, thus allowing rapid formation of the *closed* RNC•SRP•FtsY targeting complex at the membrane. Finally, the SecYEG complex drives additional NG-domain rearrangements that are coupled to GTPase activation and unloading of the RNC onto the translocon (*Shen et al., 2012*; *Akopian et al., 2013a*; *Ataide et al., 2011*). These substrate- and translocon-driven conformational changes of the SRP•FtsY complex ensure productive targeting of the correct SRP substrates, while also providing key mechanisms to reject SRP-independent proteins from the pathway (*Figure 1*).

Compared to the RNC, the interaction of FtsY with phospholipid membranes and the role of membrane in the targeting reaction remain incompletely understood. FtsY has a poorly conserved acidic A-domain preceding its NG-domain. The A-domain has been implicated in interaction of FtsY with the membrane and translocon, but its precise roles remain unclear (*de Leeuw et al., 2000*; *Angelini et al., 2005*, *Angelini et al., 2006*; *Weiche et al., 2008*; *Braig et al., 2009*; *Kuhn et al., 2015*). Most previous work has focused on an amphiphilic lipid binding helix, here termed αN1, at the junction between A- and NG-domains. These studies showed that while the FtsY NG-domain itself could not stably bind membrane nor support efficient protein targeting, inclusion of Phe196 from the A-domain stabilizes the helical structure of αN1 and restores both activities in the FtsY-NG + 1 construct (*Parlitz et al., 2007*; *Lam et al., 2010*; *Stjepanovic et al., 2011*).

Importantly, lipid interaction via αN1 strongly stimulates FtsY's GTPase activity and formation of the *closed* SRP•FtsY complex (*Lam et al., 2010*; *Stjepanovic et al., 2011*). This is because in free FtsY, αN1 tightly packs against the remainder of the N-domain and sterically occludes approach of Ffh (*Neher et al., 2008*; *Draycheva et al., 2016*). Multiple observations suggest that lipid interaction of αN1 removes it from these auto-inhibitory contacts and thus primes FtsY for complex formation with SRP: (i) αN1 was proteolytically cleaved in all the crystal structures of *closed* SRP•FtsY complexes (*Shepotinovskaya and Freymann, 2002*; *Gawronski-Salerno and Freymann, 2007*; *Shepotinovskaya et al., 2003*; *Egea et al., 2004*; *Focia et al., 2004*); (ii) EPR studies showed that formation of the *closed* complex with SRP leads to significant mobilization of the αN1 helix, suggesting that it is released from the remainder of FtsY (*Lam et al., 2010*); (iii) consistent with the prediction from (ii), closed complex formation with SRP substantially increases the binding of FtsY to liposomes (*Lam et al., 2010*; *Parlitz et al., 2007*); and (iv) an FtsY-dN1 mutant, in which αN1 is deleted, is superactive in GTPase activity and in stable complex formation with SRP (*Neher et al., 2008*), phenocopying the stimulatory effect of lipids on FtsY. Together with the finding that lipids

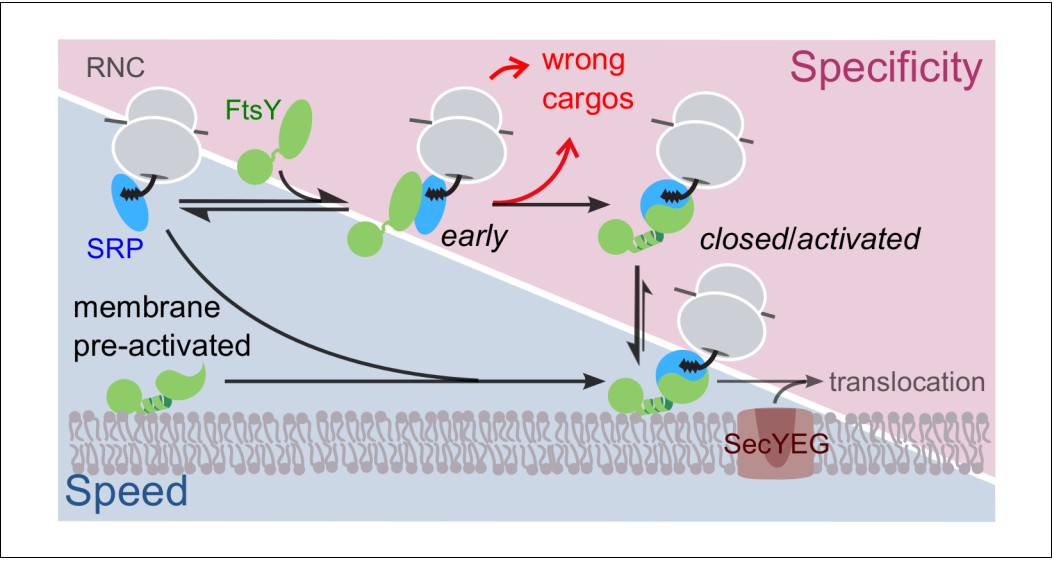

**Figure 1.** Schematic of the current models for co-translational protein targeting by the SRP pathway. Targeting via FtsY molecules that are pre-bound and activated at the membrane is shown on the lower left (shaded in *blue*); the alternative targeting route via FtsY molecules in solution is shown on the upper right (shaded in *magenta*).

are also required for the interaction of FtsY with the SecYEG translocon (*Kuhn et al., 2015*), these observations led to the current model in which most FtsY molecules are membrane-bound through the αN1 helix and pre-activated for receiving cargo-loaded SRP (*Kuhn et al., 2015*; *Parlitz et al., 2007*; *Lam et al., 2010*; *Draycheva et al., 2016*; *Braig et al., 2011*) (*Figure 1*, lower pathway in *blue*).

Nevertheless, rigorous formulation and consideration of this model lead to a number of conceptual conundrums. During SRP-dependent targeting, substrate-driven assembly of the SRP•FtsY *closed* complex is the major fidelity checkpoint at which incorrect SRP substrates are rejected, as they mediate this assembly up to $10^3$-fold more slowly than the correct substrates (*Zhang et al., 2010*). However, liposome-activated FtsY rapidly forms a closed complex with SRP even in the absence of the RNC (*Lam et al., 2010*). Thus, if the pathway occurs primarily through pre-activated FtsY at the membrane according to the current model (*Figure 1*, lower route in *blue*), this would potentially bypass an important selection mechanism in the SRP pathway. The alternative path, in which FtsY assembles a *closed* complex with SRP in the cytosol and then interacts with the membrane (*Figure 1*, upper route in *magenta*), would preserve the specificity of substrate selection, but the unfavorable pre-equilibrium for FtsY's membrane interaction and activation could render the pathway less efficient. These considerations delineate a common challenge that biological pathways face in the trade-off between efficiency and specificity; how the SRP pathway overcomes this challenge is unclear.

Indeed, many observations suggest that the lipid interaction of FtsY is more complex than depicted in the current model. The notion that FtsY contains multiple membrane binding motifs was initially suggested by the observation that FtsY induces extensive liposome aggregation, implying a single FtsY interacting with multiple membranes (*de Leeuw et al., 2000*). Later, the extreme N-terminus of the A-domain was identified as a secondary lipid binding motif based on sequence conservation and its ability to confer carbonate-resistant membrane association of FtsY (*Weiche et al., 2008*; *Braig et al., 2009*). However, the function of the additional membrane-binding motif(s) in FtsY and its relationship with the αN1 motif remain unclear.

To address these issues, we carried out the first quantitative analysis of FtsY's membrane interaction using single-molecule fluorescence microscopy on supported lipid bilayers. The sensitivity and resolution of this assay allowed us to detect two distinct modes of membrane interactions: a Dynamic mode mediated by the extreme N-terminus of the A-domain, and a Stable mode mediated by the αN1 motif. Free FtsY is auto-inhibited and interacts with the membrane primarily in the Dynamic mode. A conformational change is required for FtsY to engage the membrane in the Stable mode, which is driven by complex assembly with SRP. An engineered FtsY pre-organized into the Stable mode led to indiscriminate targeting in vitro and was unable to support cell growth in vivo. These results lead to a new model in which the Dynamic mode allows the membrane association of FtsY to be uncoupled from its conformational activation at the membrane, thus ensuring both the efficiency and fidelity of the targeting pathway.

## Results

### SRP induces a switch in the membrane interaction mode of FtsY

To rigorously investigate FtsY-membrane interactions at high resolution, we assembled supported lipid bilayers (SLB) on microscope coverslips (*Lin et al., 2010*; *Cremer and Boxer, 1999*; *Seu et al., 2007*). Total internal reflection fluorescence (TIRF) microscopy allowed us to directly observe membrane association events of fluorescently labeled FtsY on the SLB at single-molecule resolution (*Figure 2A*). Since TIRF illumination excites ~100 nm above the focal surface, the association and dissociation of individual FtsY molecules to and from the SLB were monitored by the appearance and disappearance, respectively, of quantized, discrete fluorescent spots. Time trajectories for the appearance, movement, and disappearance of individual fluorescent spots on SLB were constructed using an established particle-tracking routine (see Materials and methods). Representative trajectories are shown in *Figure 2B–D*.

We observed two types of trajectories that represent FtsY molecules interacting with the membrane in distinct modes. With free FtsY, most of the molecules associated with and dissociated from SLB rapidly, while a small fraction of molecules stably associated with and diffused two-dimensionally

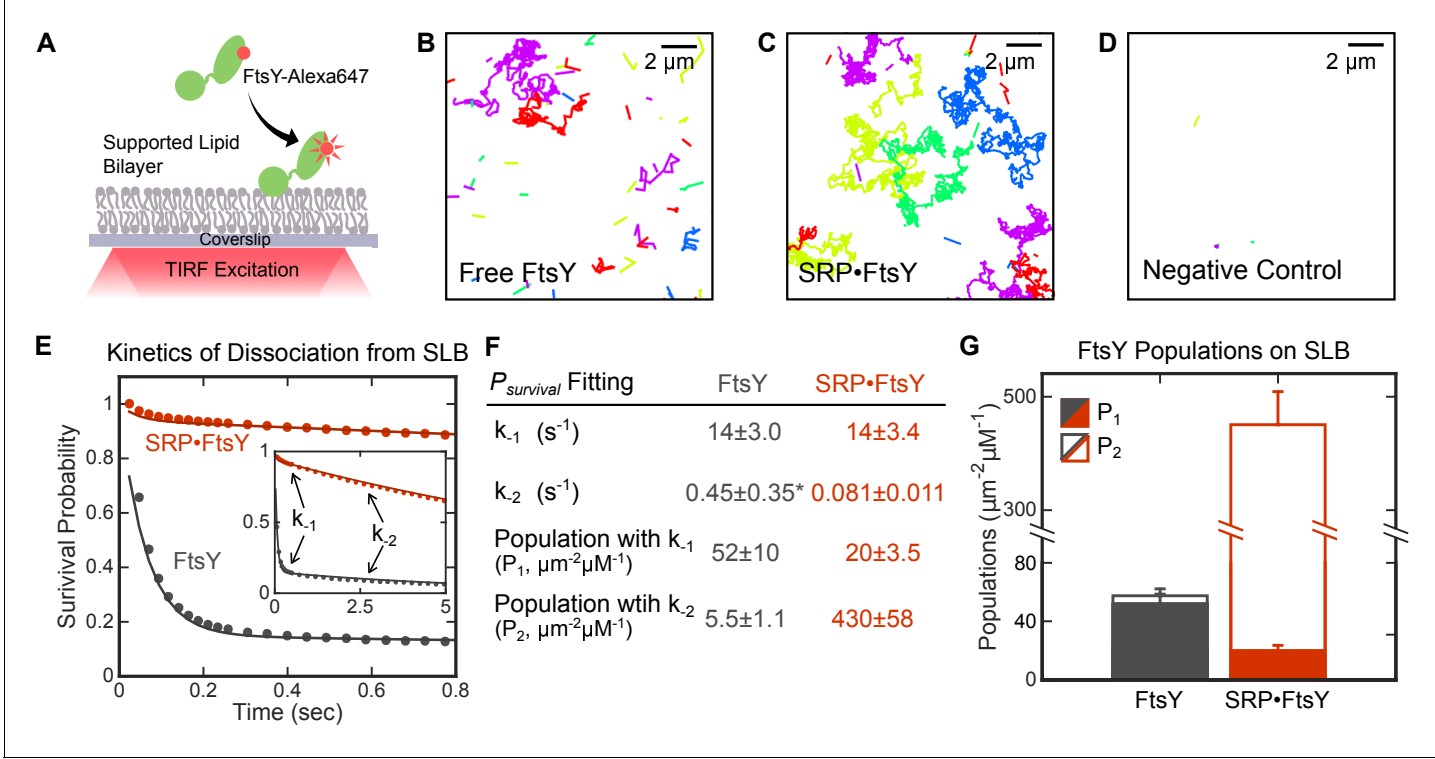

**Figure 2.** Single molecule analyses detected two distinct modes of FtsY-lipid interactions that are regulated by SRP. (**A**) Schematic of smTIRF setup for observing FtsY-membrane interaction on the SLB. FtsY was labeled with Alexa647 at position C345. (**B–D**) Representative trajectories of free FtsY molecules (**B**), SRP•FtsY complexes (**C**), and FtsY-dN1 (**D**) on SLB within an arbitrary section of 100 frames. The colors are randomly assigned to distinguish different molecules. (**E**) Representative data (dots) and fitting curves (lines) of the survival probability distribution of trajectories on SLB for free FtsY (*grey*) and the SRP•FtsY complex (*red*). The data were fit to *Equation 3*, and the obtained parameters were summarized in parts F and G. (**F**) Summary of the dissociation rate constants and population distributions obtained from the survival probability analyses in part E. *The fitting of $k_{-2}$ in free FtsY is only accurate to an order of magnitude, due to the small population of free FtsY in the Stable mode. (**G**) Summary of the population distributions in the Dynamic (filled bars) and Stable (open bars) modes in free FtsY (grey) and the SRP•FtsY complex (red). All values are reported as mean ± S.D., with n ≥ 3.

The following figure supplements are available for figure 2:

**Figure supplement 1.** Single-exponential functions do not adequately fit $P_{survival}(t)$ data.

**Figure supplement 2.** Photobleaching is slow and does not interfere with the lifetime analysis.

on the SLB (*Figure 2B* and *Video 1*). Importantly, when FtsY assembles a stable complex with SRP in the presence of the non-hydrolyzable GTP analog GppNHp (5'-Guanylyl imidodiphosphate), most of the SRP•FtsY complexes stably bound to membrane, while a small fraction of the complex exhibited rapid association and dissociation on the SLB (*Figure 2C* and *Video 2*). Few fluorescence spots were detected on the SLB with a negative control FtsY-dN1, in which all the potential membrane interacting motifs in FtsY were deleted (see Introduction and Figure 5 below), indicating low false-positive signals from free dye or background noise (*Figure 2D* and *Video 3*).

To quantify the dissociation rate constants of FtsY from SLB, we calculated the Survival Probability Distribution, $P_{survival}(t)$, for all the trajectories under each condition. $P_{survival}(t)$ is defined as the probability that individual particles remain on the SLB for another time interval $t$, given that the particle is initially on the SLB (see Materials and methods). The $P_{survival}(t)$ curves in the absence and presence of SRP were both bi-phasic and fit well to the sum of two exponential functions with dissociation rate constants of $k_{-1} = 14$ s$^{-1}$ and $k_{-2} = 0.081$ s$^{-1}$ (*Figure 2E and F*), whereas single exponential functions do not adequately fit the data (*Figure 2—figure supplement 1*). This corroborates the presence of two populations of FtsY or SRP•FtsY complexes that interact with the membrane with distinct kinetic

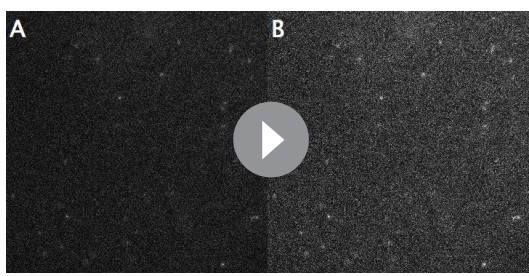

**Video 1.** Interaction of Free FtsY with SLB. A representative movie of free FtsY interacting with SLB is shown as raw data in (A) and processed data with colored trajectories in (B). The frame speed is slowed down by 2-fold (~40 ms/frame) for better visualization. Colors were randomly assigned to distinguish different molecules.

stabilities. The rate constants for individual populations are the same, within experimental error, for free FtsY and the SRP•FtsY complex (*Figure 2F*). While >90% of free FtsY are in the rapidly-dissociating population, the slowly-dissociating population dominates the SRP•FtsY complex (*Figure 2F and G*, P$_1$ and P$_2$).

These data support a model in which FtsY samples two conformations that interact with the membrane in distinct modes, termed the 'Dynamic' and 'Stable' modes for the rapidly- and slowly-dissociating populations, respectively, and SRP binding shifts the conformational equilibrium of FtsY towards the Stable mode (*Figure 3A and B*). Although more complex models can be invoked, there was no evidence for additional conformational states of FtsY that affect its membrane interaction; thus, *Figure 3* depicts the simplest model that accounts for all the SLB data in this study. The remainder of the rate and equilibrium constants in this model were determined or calculated as follows.

To analyze the kinetics of FtsY association with SLB, we quantified the cumulative appearances of new SLB-bound trajectories as a function of time (*Figure 3C and D*). As the lifetime of the trajectory on SLB (τ) for the Dynamic mode is >100 fold shorter than the Stable mode, we used a cutoff of τ = 0.25 s to distinguish trajectories in the Dynamic and Stable modes; use of this cutoff gave population quantifications that agreed well with the results from survival probability analysis (cf. *Figure 3—figure supplement 1* versus *Figure 2F*, P$_1$ and P$_2$). Linear fits of the data gave the apparent association rate constant for molecules that bind the membrane in each mode (*Figure 3C and D* and *Figure 3—figure supplement 1*, $k_{1,app}$ and $k_{2,app}$, respectively). These association rate constants are apparent, as the number of accumulated traces were normalized by the total concentration of FtsY or SRP•FtsY complex and did not take into account the fraction of molecules in each conformation in solution. Numerically solving the models in *Figure 3A and B* based on mass conservation and the measured kinetic parameters allowed us to extract true FtsY-membrane association rate constants in the Dynamic and Stable modes (*Figure 3E*, $k_1$ and $k_2$, respectively; see Materials and methods). The conformational equilibria between the Dynamic and Stable modes at the membrane ($K_{mem}^{FtsY}$ and $K_{mem}^{SRP•FtsY}$ in *Figure 3A and B*, respectively) were directly obtained from the ratios of membrane-bound populations in the two

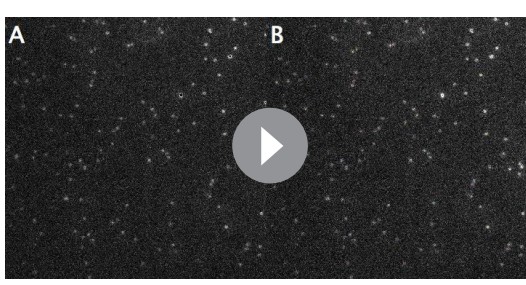

**Video 2.** Interaction of the SRP•FtsY complex with SLB. A representative movie of SRP•FtsY complexes interacting with SLB is shown as raw data in (A) and processed data with colored trajectories in (B). The frame speed is slowed down by 2-fold (~40 ms/frame) for better visualization. Colors were randomly assigned to distinguish different molecules.

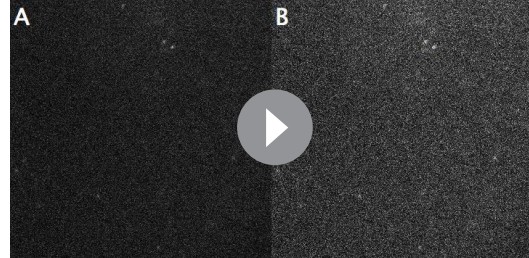

**Video 3.** Interaction of the SRP•FtsY-dN1 complex with SLB, as negative control. A representative movie of SRP•FtsY-dN1 associating with SLB is shown as raw data in (A) and as processed data with colored trajectories in (B). The frame speed is slowed down by 2-fold (~40 ms/frame) for better visualization. Colors were randomly assigned to distinguish different molecules.

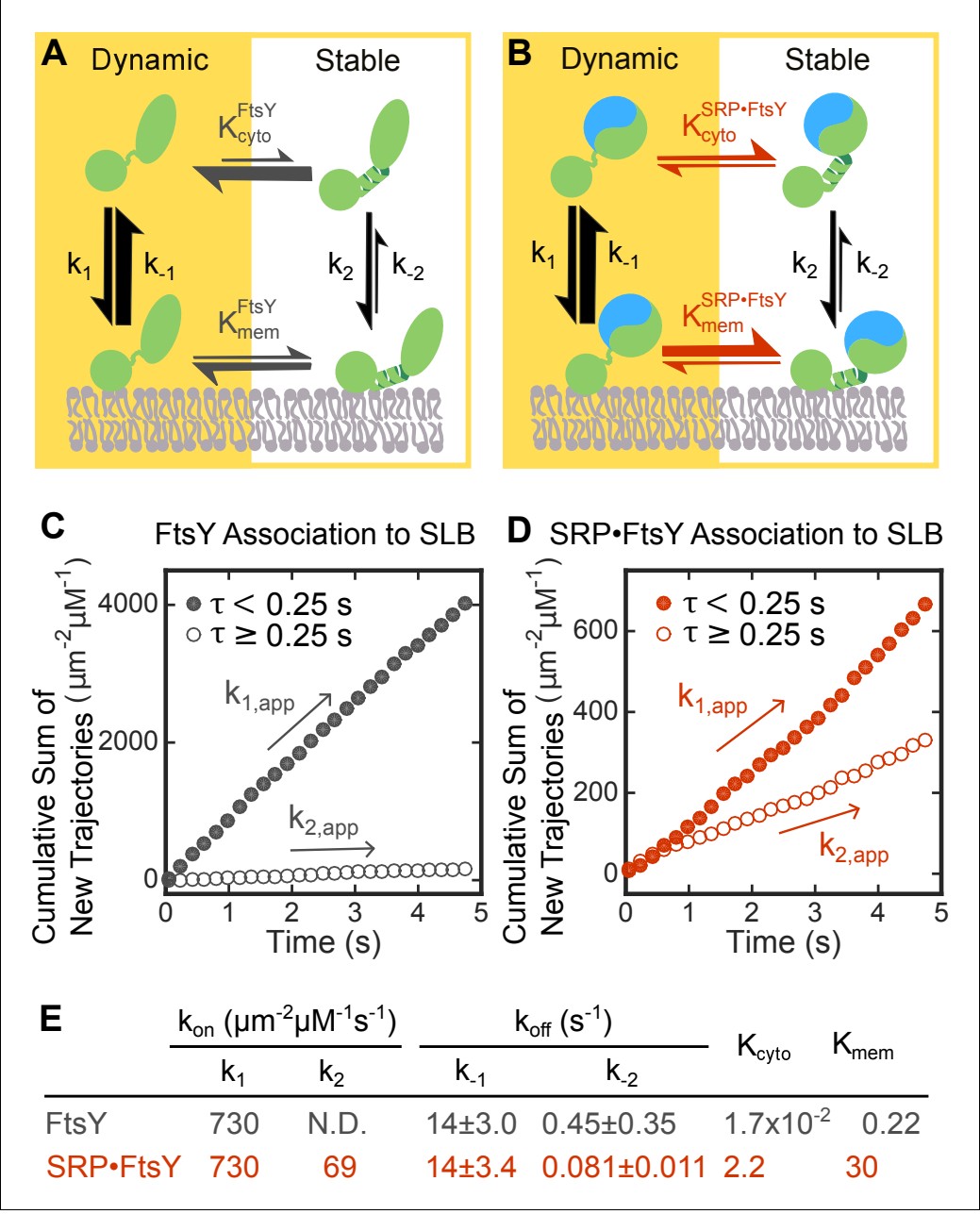

**Figure 3.** SRP binding drives FtsY from the Dynamic to the Stable mode. (A, B) Thermodynamic models of FtsY-membrane interaction in free FtsY (A) and the SRP•FtsY complex (B). (C, D) Representative data for apparent association kinetics of FtsY (C) and the SRP•FtsY complex (D) with SLB in the Dynamic (closed circles) and Stable mode (open circles). (E) List of the rate and equilibrium constants for the models in parts A and B. N.D., not determined with confidence due to the unstable fitting of $k_{-2}$. Values are reported as mean ± S.D., with n ≥ 3.
The following figure supplement is available for figure 3:

**Figure supplement 1.** Summary of the population distributions and apparent association rate constants obtained by using a lifetime cutoff of 0.25 s to distinguish trajectories in the Dynamic and Stable modes.

modes ($P_1$ and $P_2$ in *Figure 2F*). The conformational equilibria in solution ($K_{cyto}^{FtsY}$ and $K_{cyto}^{SRP•FtsY}$) were calculated based on the measured kinetic parameters and thermodynamic coupling of the conformational equilibria to the equilibria of FtsY-membrane interactions (see Materials and methods).

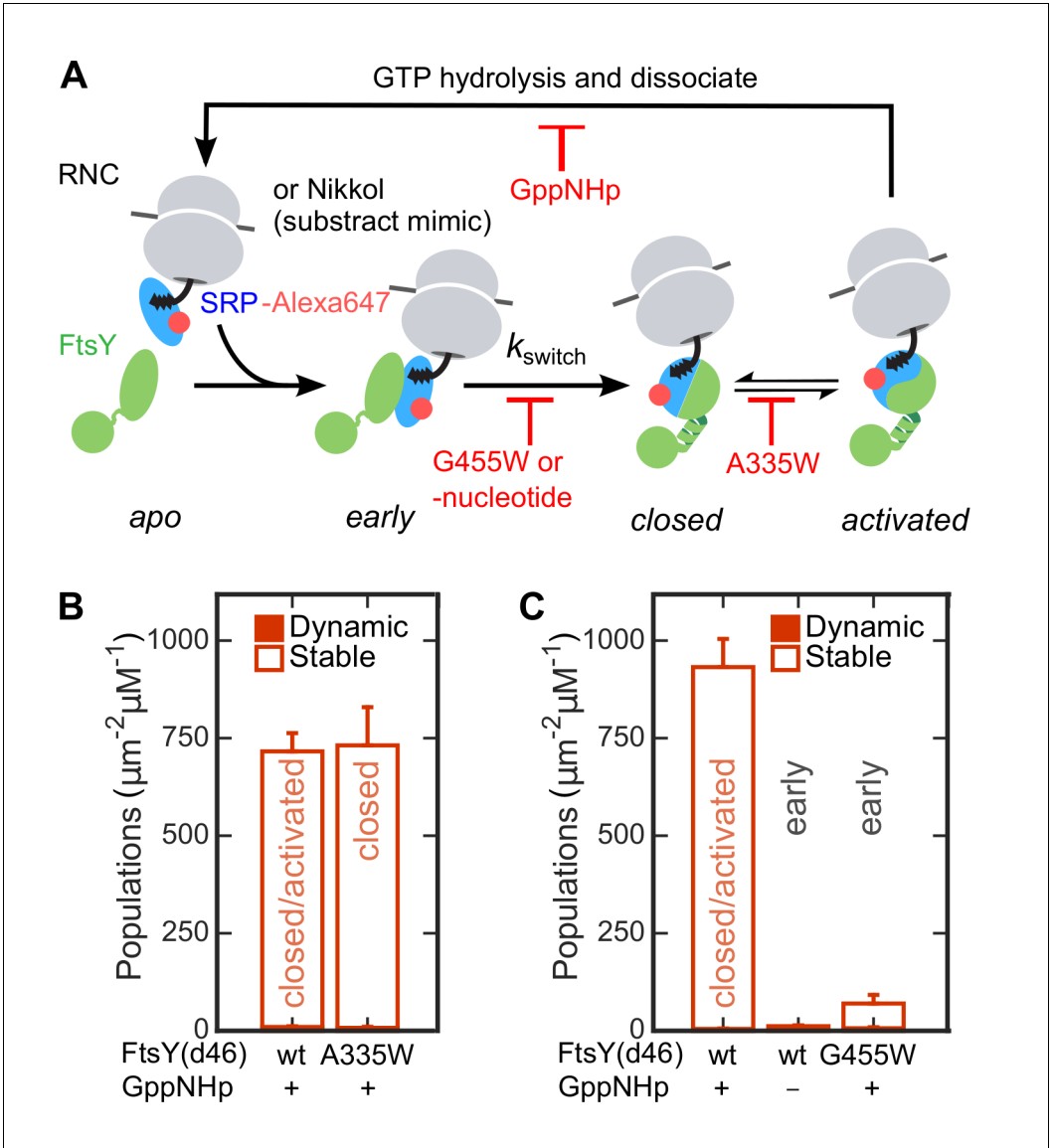

**Figure 4.** The Dynamic-to-Stable transition occurs during the *early*-to-*closed* rearrangement in the SRP•FtsY complex. (**A**) Schematic of conformational changes in the SRP-FtsY GTPase cycle and the conditions that stall the complex at different conformational stages. SRP was labeled with Alexa647 at C153. (**B, C**) Comparison of the lipid interactions of the SRP•FtsY complex with wildtype proteins in GppNHp (a mixture of *closed/activated* states) and with the complex stalled in the *closed* state (**B**) or the *early* state (**C**). Filled and open bars represent populations in the Dynamic and Stable modes, respectively, determined using the lifetime cutoff of 0.25 s. As expression of full-length FtsY(A335W) is toxic to the cell, FtsY-d46 was used for the measurements in these experiments. Except for the lipid interactions, FtsY-d46 behaves identically to full-length FtsY in the SRP/FtsY GTPase cycle and in activation by RNC (*Powers and Walter, 1997*; *Zhang et al., 2009*). A substrate mimic, Nikkol (*Bradshaw et al., 2009*), was included in (**B**) to facilitate complex formation. RNC$_{FtsQ}$ was included in (**C**) to stabilize the *early* complex. Values are reported as mean ± S.D., with n $\geq$ 3.

Inspection of the parameters in this model yielded three interesting observations. First, FtsY-membrane association is ~10 fold slower in the Stable than the Dynamic mode (*Figure 3E*, $k_1 / k_2$). Second, SRP drives the conformational equilibrium of FtsY to the Stable mode ~130 fold ($K_{cyto}^{SRP•FtsY}$ / $K_{cyto}^{FtsY}$ and $K_{mem}^{SRP•FtsY}$ / $K_{mem}^{FtsY}$). Finally, the Dynamic-to-Stable transition of FtsY is favored at least 14-fold at the membrane ($K_{mem}$ / $K_{cyto}$). Together, these observations provide the first quantitative

evidence for a regulatory switch at FtsY's membrane recruitment step, during which FtsY tunes its membrane interactions in response to SRP.

To identify the conformational change responsible for the SRP-induced switch of FtsY from the Dynamic to the Stable mode, we used a previously characterized set of mutant GTPases or GTP analogues that stall the SRP•FtsY complex at distinct conformational stages. Omission of nucleotide or mutant FtsY(G455W) inhibits the *early*-to-*closed* rearrangement, and thus locks the SRP•FtsY complex at the *early* intermediate stage (*Figure 4A*, [*Zhang et al., 2008*]). Mutant FtsY(A335W) inhibits active site rearrangements that lead to GTPase activation, and thus locks the SRP•FtsY complex in the *closed* state (*Figure 4A*, [*Shan et al., 2004*]). We tested the membrane-binding abilities of SRP•FtsY complexes assembled with these mutants or nucleotide analogues using the SLB-smTIRF setup. As excess FtsY was needed to drive formation of the *early* complex, we labeled SRP instead of FtsY in these experiments to remove contributions from free FtsY. SRP•FtsY(A335W) displayed comparable amounts of Stable SLB interactions as the wildtype complex (*Figure 4B*). In contrast, both conditions that stall the SRP•FtsY complex in the *early* conformational state significantly reduced the Stable mode of lipid interaction (*Figure 4C*). Thus, FtsY switches to the Stable mode during the *early*-to-*closed* rearrangement in the SRP•FtsY complex.

Together, the results in this section show that FtsY interacts with the membrane in two modes, a Dynamic and a Stable mode. While free FtsY predominantly interacts with the membrane in the Dynamic mode, complex formation with SRP drives most of FtsY into the Stable mode of interactions. The Dynamic-to-Stable switch occurs during the rearrangement of the SRP•FtsY complex from the *early* to the *closed* conformation.

## Two functionally important motifs in FtsY mediate the Dynamic and Stable modes

To define the sites responsible for FtsY's Dynamic and Stable interactions, we constructed a series of FtsY mutants in which individual motifs and domains of FtsY are systematically truncated (*Figure 5A*). FtsY-d14 removes the N-terminal 14 residues of FtsY (termed the αA1 motif), which has been proposed as a secondary lipid binding motif (*Weiche et al., 2008*; *Braig et al., 2009*). FtsY-d46 removes a more substantial portion of the A-domain. FtsY-NG+1 and FtsY-NG contain the well-characterized αN1 helix, and the additional Phe196 in FtsY-NG+1 stabilizes this helix (*Parlitz et al., 2007*). Finally, FtsY-dN1 provides a negative control in which the entire A-domain and αN1 helix are deleted. Measurement of GTPase activity (*Peluso et al., 2001*) showed that in the absence of membrane, all of these mutants behave identically to full-length FtsY in association and reciprocal GTPase activation with SRP (*Figure 5—figure supplement 1* and [*Powers and Walter, 1997*; *Shan et al., 2007*; *Bahari et al., 2007*]).

We measured the interactions of mutant FtsYs with SLB using smTIRF. Since the Dynamic and Stable modes dominate free FtsY and the SRP•FtsY complex, respectively, we tested these mutants under both conditions to dissect the contributions of potential binding motifs to each interaction mode. Representative trajectories are shown in *Figure 5—figure supplement 2*. With free FtsY, removal of the αA1 motif (FtsY-d14) reduced interactions in the Dynamic mode >10 fold, and no substantial additional reductions were observed with further truncations (*Figure 5B*, solid bars). This indicates that the αA1 motif is primarily responsible for FtsY-lipid interactions in the Dynamic mode. In contrast, only complete truncation of the αN1 helix in FtsY-dN1 abolishes the Stable mode of interactions in the SRP•FtsY complex (*Figure 5C*, open bars), indicating that the αN1 motif mediates the Stable mode. Thus, distinct motifs in FtsY mediate the Dynamic and Stable modes of lipid interactions.

To further probe the nature of the Dynamic mode, we tested the contribution of conserved basic residues in the αA1 motif (Lys3, Lys5, Lys6 and Arg7) (*Figure 5D*, [*Weiche et al., 2008*]); basic residues are enriched in αA1 and could mediate interaction with the anionic phospholipid headgroup (*de Leeuw et al., 2000*; *Lam et al., 2010*). To this end, we constructed FtsY mutants in which part or all of these charges are removed and/or reversed (*Figure 5D*, EE, EL, and EEEL). All these mutants exhibited two-fold reduced interactions with the membrane compared to wildtype FtsY in SLB-smTIRF measurements (*Figure 5E*). Nevertheless, the defects of these mutants are modest compared to FtsY-d14 (*Figure 5E*), suggesting that the aromatic and aliphatic residues in the remainder of αA1 also contribute to interaction in the Dynamic mode. Thus, the Dynamic mode is driven by a combination of electrostatic and hydrophobic interactions mediated by αA1.

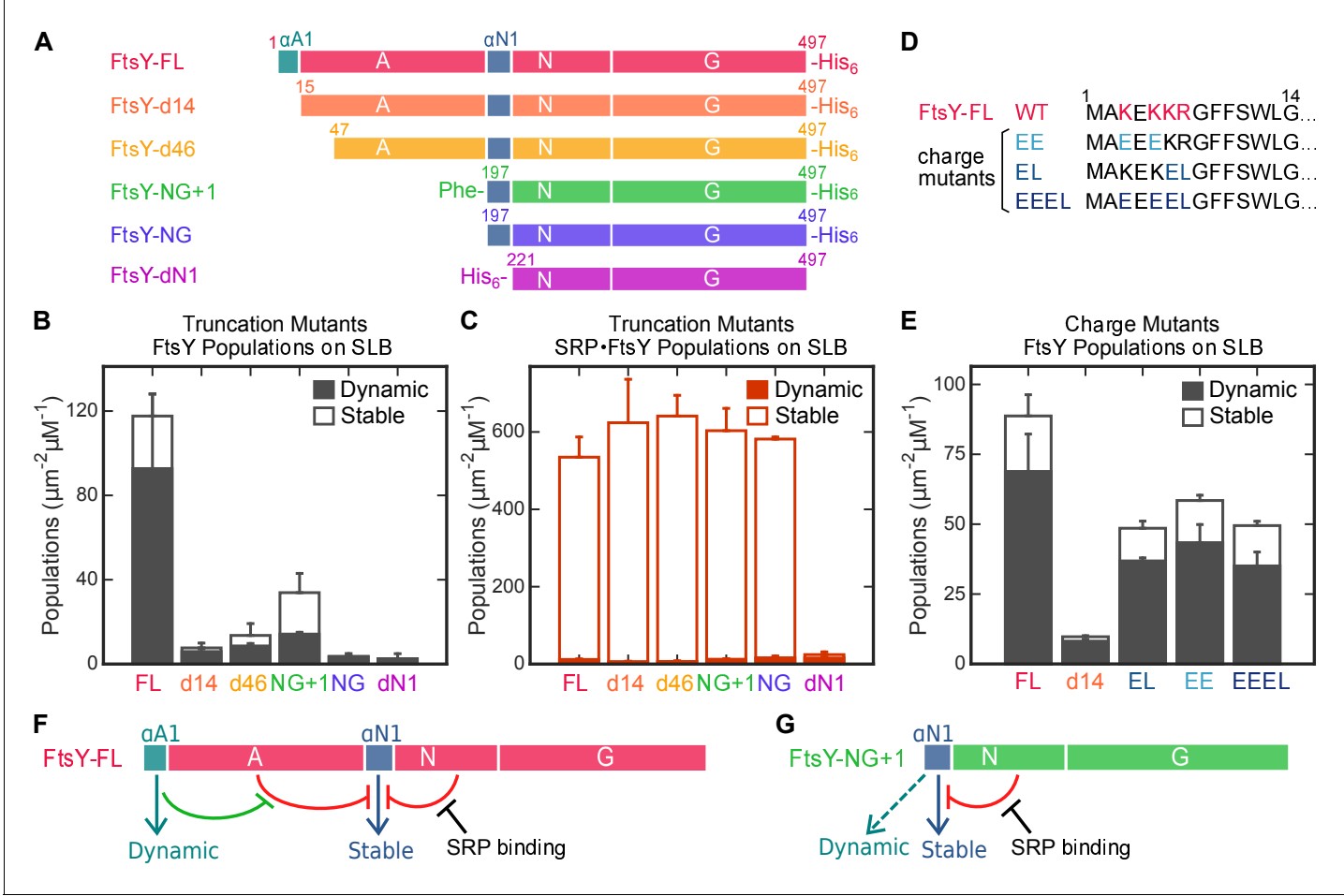

**Figure 5.** Distinct motifs in FtsY mediate the two membrane interaction modes. (**A**) Domain structures of wildtype FtsY and FtsY truncation mutants. (**B, C**) The membrane binding abilities of the truncation mutants on SLB as free FtsY (**B**) and SRP•FtsY complex (**C**). Filled and open bars represent populations in the Dynamic and Stable modes, respectively, determined using the lifetime cutoff of 0.25 s. (**D**) Sequences of the αA1 motif in wildtype and charge mutants. Charged residues in wildtype sequence are highlighted in *red*, and their mutations are highlighted in *blue*. (**E**) Charge mutations in αA1 reduced membrane interactions of free FtsY with SLB. (**F, G**) Model for regulation of the membrane interactions of full length FtsY (**F**) and FtsY-NG+1 (**G**). Cyan and blue arrows denote membrane interactions in the Dynamic and Stable modes, respectively. Both the A-domain (this work) and N-domain (*Parlitz et al., 2007*; *Lam et al., 2010*) inhibit FtsY from engaging the membrane in the Stable mode (red inhibition marks). In full-length FtsY, αA1 partially relieves the inhibition from the A-domain (green inhibition mark). In FtsY-NG+1, the αN1 motif can also mediate some degree of Dynamic interactions (dashed arrow in **G**). With both constructs, interaction with SRP is the most effective mechanism to relieve the inhibitory effect from the N-domain and allow FtsY to interact with the membrane in the Stable mode. Values are reported as mean ± S.D., with n ≥ 3.

The following figure supplements are available for figure 5:

**Figure supplement 1.** FtsY truncation mutants exhibit no defects in complex formation and GTPase activation with SRP in the absence of lipids.

**Figure supplement 2.** Representative trajectories of free FtsY molecules (**A**) and SRP•FtsY complexes (**B**) on SLB for wildtype and truncation mutants of FtsY.

**Figure supplement 3.** Lipid activation of FtsY's GTPase activity independently probes the ability of free FtsY to sample the Stable mode.

**Figure supplement 4.** The interaction of FtsY with SLB is insensitive to the identity of anionic phospholipids.

Importantly, FtsY-d14 not only reduced the Dynamic mode, but also nearly abolished the Stable mode of interaction in free FtsY (*Figure 5B and E*), suggesting that interaction in the Dynamic mode is required to attain the Stable mode in free FtsY. Consistent with this notion, disruption of the Dynamic and Stable interactions are highly correlated in the αA1 charge mutants (*Figure 5E*). To independently test this model, we measured the lipid-mediated stimulation of FtsY's GTPase activity, which provides a readout for the conformational activation of FtsY upon engagement of the αN1 helix with the membrane (*Lam et al., 2010*). The GTPase activity of wildtype FtsY is strongly stimulated by PG/PE liposomes as previously observed, whereas this stimulation was abolished in FtsY-d14, similar to the behavior of FtsY-NG (*Figure 5—figure supplement 3*). Together, these results strongly suggest that in free FtsY, the Dynamic mode is upstream of and required for this receptor to further engage the membrane in the Stable mode.

To assess the contribution of the Dynamic interactions to biological function, we tested the ability of FtsY mutants to mediate the co-translational targeting and translocation of a model SRP substrate, preprolactin (pPL), into ER microsomal membranes (see Materials and methods; *Figure 6—figure supplement 1*). Removal of the αA1 motif reduced targeting efficiency to less than 30% of wildtype level (*Figure 6A*, FtsY-d14), indicating that αA1 is important for protein targeting. The set of αA1 charge mutants of FtsY (*Figure 5D*) provided a more controlled assessment of the contribution of the Dynamic mode. All these mutants reduced the efficiency of pPL targeting and translocation (*Figure 6A*), and the targeting efficiency correlated well with the amount of Dynamic interactions displayed by each variant in this set of mutants (*Figure 6B*). These results provide strong evidence for an important role of the Dynamic interactions in protein targeting mediated by full-length FtsY.

Intriguingly, as the remainder of the A-domain was further truncated in d14, d46 and NG+1, the Stable mode of lipid interactions in free FtsY (*Figure 5B*, open bars) was gradually restored. This suggests that the A-domain prevents free FtsY from engaging with the membrane in the Stable mode (*Figure 5F*, red line from A-domain). With FtsY-NG+1, in which the entire A-domain is removed except for Phe196, the Stable and Dynamic interactions were restored to ~80% and ~20% of that of full-length FtsY, respectively (*Figure 5B*). This suggests that when the inhibitory A-domain was removed, the αN1 helix could also mediate Dynamic interactions with the membrane, albeit less efficiently than the αA1 motif (*Figure 5G*, dashed green arrow). As αN1 also mediates the Stable interaction, transition from the Dynamic to the Stable mode is likely more efficient in FtsY-NG+1 than in full-length FtsY. Nevertheless, FtsY-NG+1 still interacts weakly with the membrane by itself and requires SRP binding to drive favorable interaction with the membrane (cf. *Figure 5B* vs. *Figure 5C*). Thus, additional auto-inhibitory interactions, presumably from the remainder of the N-domain (*Neher et al., 2008*; *Draycheva et al., 2016*), are present to prevent free FtsY-NG+1 from engaging with the membrane in the Stable mode (*Figure 5F and G*, red lines from N-domain). Finally, protein targeting efficiency paralleled the restoration of the Stable mode in free FtsY across the A-domain truncation mutants (*Figure 6C and D*), consistent with the Stable mode being an obligatory species during targeting.

Collectively, the results in this section showed that the Dynamic and Stable modes are mediated primarily by the αA1 and αN1 motifs of FtsY, respectively (*Figure 5F*, cyan and blue arrows). Before association with SRP, lipid interaction via αA1 is required for wildtype FtsY to further engage the membrane via the αN1 helix. This requirement is due, in part, to the inhibitory effect from the remainder of the A-domain, whose highly acidic nature likely repels FtsY from the membrane. The αA1 motif likely provides an initial membrane attachment to overcome this inhibition, enabling the subsequent Stable mode and efficient targeting (*Figure 5F*, green lines); this explains why αA1 can be bypassed by truncations of the A-domain (*Figure 5G*). Finally, regardless of the presence of the A-domain, free FtsY exists in an auto-inhibited state for interaction with the membrane via αN1, and interaction with SRP is the dominant mechanism to drive FtsY to the conformation that interacts with the membrane in the Stable mode (*Figure 5F and G*).

The SRP-induced transition of FtsY's membrane interaction mode is unlikely to be an artifact of the DOPC/DOPS composition of the SLB. First, bacterial SRP and FtsY can replace their mammalian homologues and mediate efficient targeting and insertion of mammalian substrates into ER microsomes (*Powers and Walter, 1997*) with a similar substrate selection pattern (*Zhang and Shan, 2012*). These observations indicate that the core regulatory mechanisms of SRP and the SRP receptor are insensitive to the difference in lipid composition between the bacterial plasma membrane

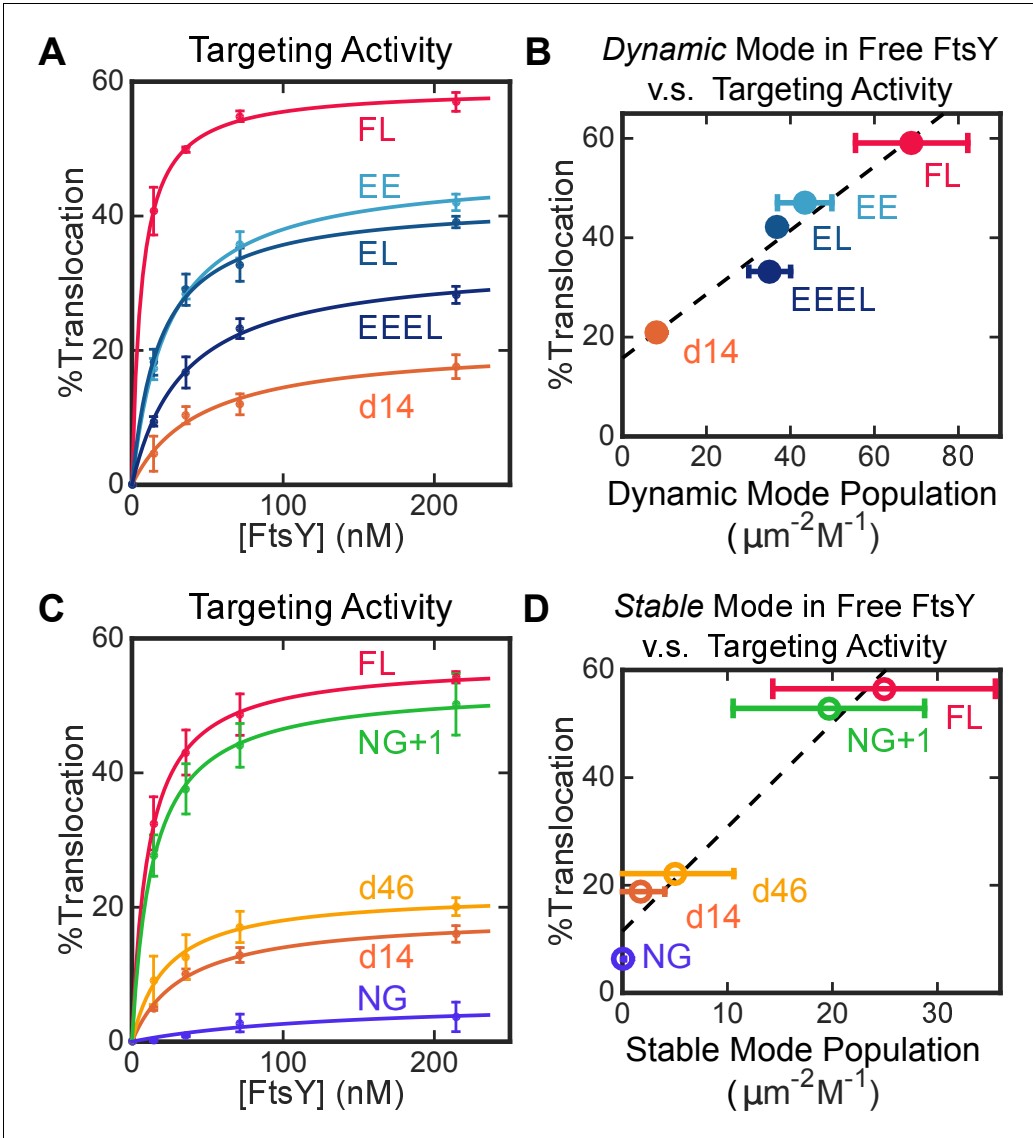

**Figure 6.** Both modes of FtsY-membrane interactions impact efficient protein targeting. (**A**) The effect of charge mutations in the αA1 motif on the co-translational targeting of pPL. Data were fitted to *Equation 7*. (**B**) Targeting efficiencies of the αA1 charge mutants correlate with the populations of molecules in the Dynamic mode in free FtsY. (**C**) The effect of A-domain truncations on the co-translational targeting of pPL. (**D**) Targeting efficiencies of A-domain truncation mutants parallel their abilities to sample the Stable mode prior to SRP binding. Values are reported as mean ± S.D., with n ≥ 3.
The following figure supplement is available for figure 6:

**Figure supplement 1.** Co-translational protein targeting and translocation by FtsY mutants.

and the mammalian ER membrane. Second, although FtsY exhibits a preference for PG, previously observed lipid-stimulation of FtsY's biochemical activities required liposomes containing 70–100% PG, whereas total *E. coli* lipids or liposome compositions that mimic the *E. coli* plasma membrane did not induce detectable stimulations of FtsY (*de Leeuw et al., 2000*; *Lam et al., 2010*; *Stjepanovic et al., 2011*). These earlier results agree well with our model, showing that the majority of FtsY is in the auto-inhibited state in its native lipid environment. Third, we repeated the SLB measurements using DOPC/DOPG side-by-side with DOPC/DOPS (*Figure 5—figure supplement 4*). No significant differences between SLBs generated from the different lipid compositions were observed,

indicating that the identity of anionic phospholipids (PG in bacterial plasma membrane and PS in the ER membrane) does not impact FtsY's lipid interaction modes. Finally, the model derived from the SLB measurements were extensively tested by in vitro targeting assays using ER microsomes and in vivo complementation assays in bacteria (see the next section); the good agreement between the results of all three assays further suggest that modest variations in lipid composition does not alter the regulatory mechanism of SRP receptor shown here.

## The Dynamic mode balances the specificity and efficiency of protein targeting

The observation of extensive auto-inhibitory mechanisms for FtsY's membrane interaction raises a fundamental question: what is the role of these auto-inhibitory interactions? What is the penalty for evolving a receptor molecule that is pre-organized into the Stable mode at the membrane? A hypothesis stems from the consideration that FtsY-membrane interaction via αN1 is coupled to conformational activation of this receptor, which enables FtsY to adopt the *closed* conformation and bind SRP much more rapidly (*Neher et al., 2008*). As discussed in the Introduction, formation of the *closed* SRP•FtsY complex is a major cargo selection step in the SRP pathway (*Zhang et al., 2010*; *von Loeffelholz et al., 2013*). FtsY molecules pre-organized into the Stable mode could potentially bypass this key checkpoint and thereby compromise fidelity. In this context, the Dynamic mode provides a mechanism for FtsY to associate with the membrane without conformational activation, and thus preserves this fidelity checkpoint.

To test this hypothesis, we engineered an FtsY that bypasses the Dynamic mode and is pre-organized into the Stable mode. We used FtsY-dN1 to mimic the effect of lipids on pre-organizing FtsY into the *closed/activated* conformation. Multiple observations support the choice of this construct: (i) in apo-FtsY, the αN1 helix sterically occludes tight SRP•FtsY association in the *closed* conformation (*Neher et al., 2008*; *Draycheva et al., 2016*; *Shepotinovskaya and Freymann, 2002*); (ii) lipid binding releases αN1 from the remainder of the protein and preorganizes FtsY into the *closed* state (*Lam et al., 2010*; *Stjepanovic et al., 2011*); (iii) as predicted from (i) and (ii), FtsY-dN1 phenocopies the effects of lipids on enhancing SRP•FtsY assembly and GTPase activation (*Neher et al., 2008*). Note that FtsY-NG+1 is not a proper construct to mimic a pre-organized FtsY, as free FtsY-NG+1 is still auto-inhibited and requires SRP to switch to the Stable mode (*Figure 5B and C*). To re-establish stable membrane association of FtsY, we tethered His$_6$-tagged FtsY-dN1 on the SLB doped with Ni-NTA-DGS lipids for in vitro assays (*Figure 7B*), or fused FtsY-dN1 to the spontaneous membrane-inserting 3L-Pf3 sequence (*Figure 8A* [*Lim et al., 2013*]) for in vivo assays.

To test whether the pre-organized FtsY can distinguish SRPs loaded with correct and incorrect cargos, we monitored the membrane targeting of RNC•SRP•FtsY complexes in the SLB-smTIRF setup (*Figure 7*). We presented RNC•SRP complexes, labeled at SRP, to either wildtype (*Figure 7A*) or pre-organized FtsY (*Figure 7B*) and monitored the appearance of membrane-bound targeting complexes in real time. We tested RNCs bearing two representative nascent chains: FtsQ, a bona-fide SRP substrate, and Luc (luciferase), a cytosolic protein. Both wildtype and engineered FtsYs efficiently targeted RNC$_{FtsQ}$ (*Figure 7A and B*, orange lines). However, while wildtype FtsY strongly rejected RNC$_{Luc}$, significant amounts of RNC$_{Luc}$ were localized to the membrane by pre-organized FtsY-dN1 (*Figure 7A and B*, blue lines). We confirmed that the observed SRP-RNC$_{FtsQ}$ targeting is dependent on FtsY (*Figure 7—figure supplement 1*). To more specifically test the role of conformational pre-organization in FtsY-dN1, we measured other FtsY A-domain truncation mutants (d14, NG +1, and NG) that showed SRP-dependent transition from the Dynamic to Stable mode. We tethered these mutants on SLB via Ni-His$_6$ interaction and compared the targeted SRP-RNC populations mediated by these constructs to those by FtsY-dN1 (*Figure 7C and D*). All these surface-tethered FtsY mutants mediated efficient targeting of RNC$_{FtsQ}$, but not RNC$_{Luc}$, to the SLB (*Figure 7C*, yellow vs. blue bars). Importantly, they all showed 5–10 fold more selective targeting than FtsY-dN1 (*Figure 7D*). Thus, pre-organizing FtsY into the Stable mode compromises its ability to reject incorrect cargos bearing SRP-independent substrates.

To examine the consequence of pre-activating FtsY in vivo, we tested the ability of the pre-organized FtsY to complement FtsY depletion and support cell growth (*Figure 8*). TM-FtsYdN1 was expressed from the pTlac18 plasmid in *E. coli* strain IY28, in which expression of chromosomal FtsY is under control of the *ara* promoter (*Bahari et al., 2007*). We tested cell growth on LB plates supplemented with either IPTG (Isopropyl β-D-1-thiogalactopyranoside) or L-arabinose after serial

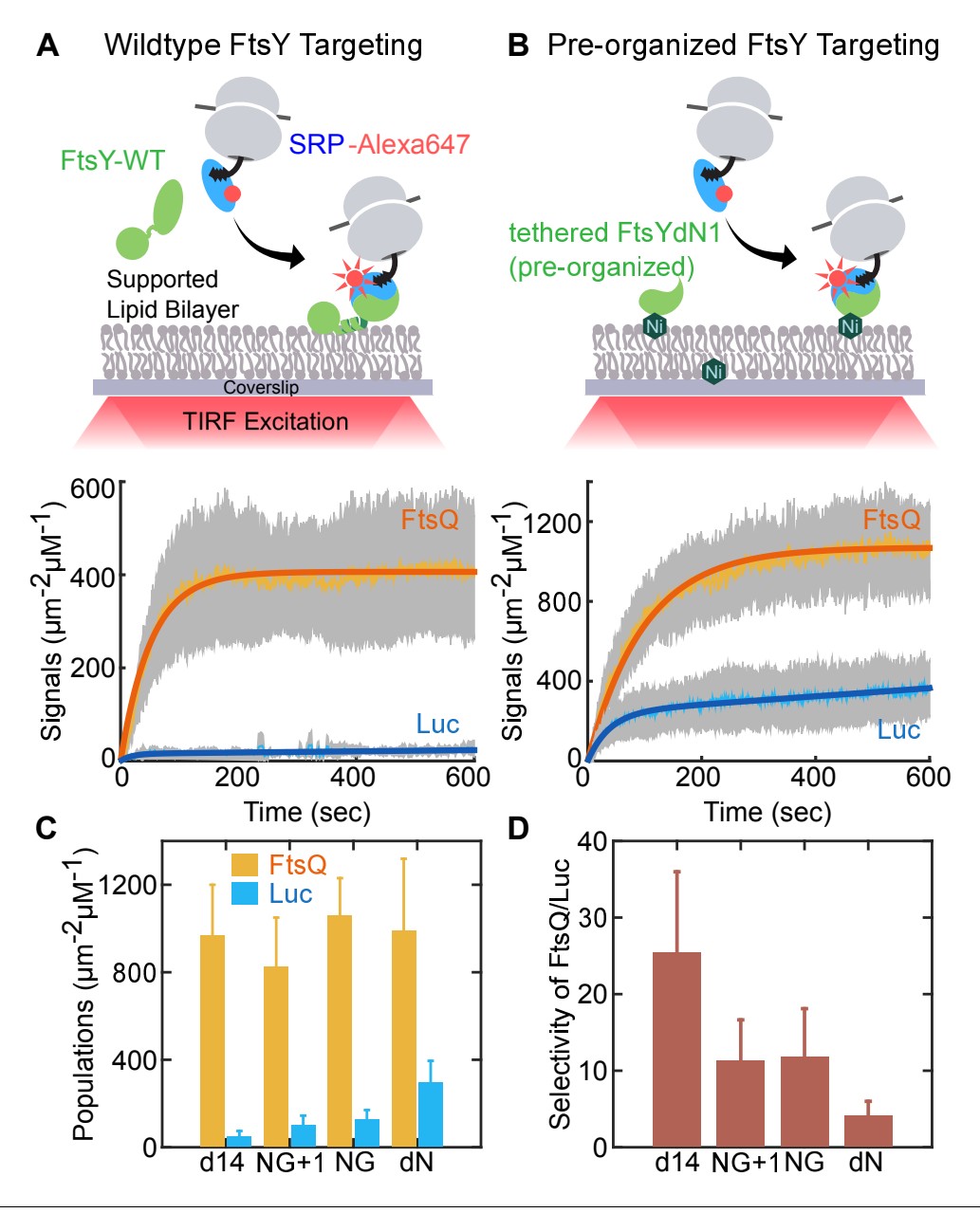

**Figure 7.** An engineered FtsY pre-organized into the Stable mode led to indiscriminate targeting. (**A**, **B**) Time courses for targeting of RNC$_{FtsQ}$ (*orange*) and RNC$_{Luc}$ (*blue*) to SLB mediated by wildtype FtsY (**A**) and FtsY pre-organized into the Stable mode (**B**). Schematics of the single-molecule real-time targeting assay is depicted above. The amounts of FtsY in the two experiments were equalized by adjusting the surface density of tethered FtsY-dN1. (**C**) The amount of RNC$_{FtsQ}$ (orange bars) and RNC$_{Luc}$ (blue bars) targeted to SLB by tethered mutant FtsYs. (**D**) Targeting specificities of SLB-tethered FtsY mutants, defined by the ratio of targeted RNC$_{FtsQ}$ over RNC$_{Luc}$ in (**C**). Values are reported as mean ± S.D., with n ≥ 3.

The following figure supplement is available for figure 7:

**Figure supplement 1.** The targeting of RNC$_{FtsQ}$•SRP to SLB is dependent on FtsY.

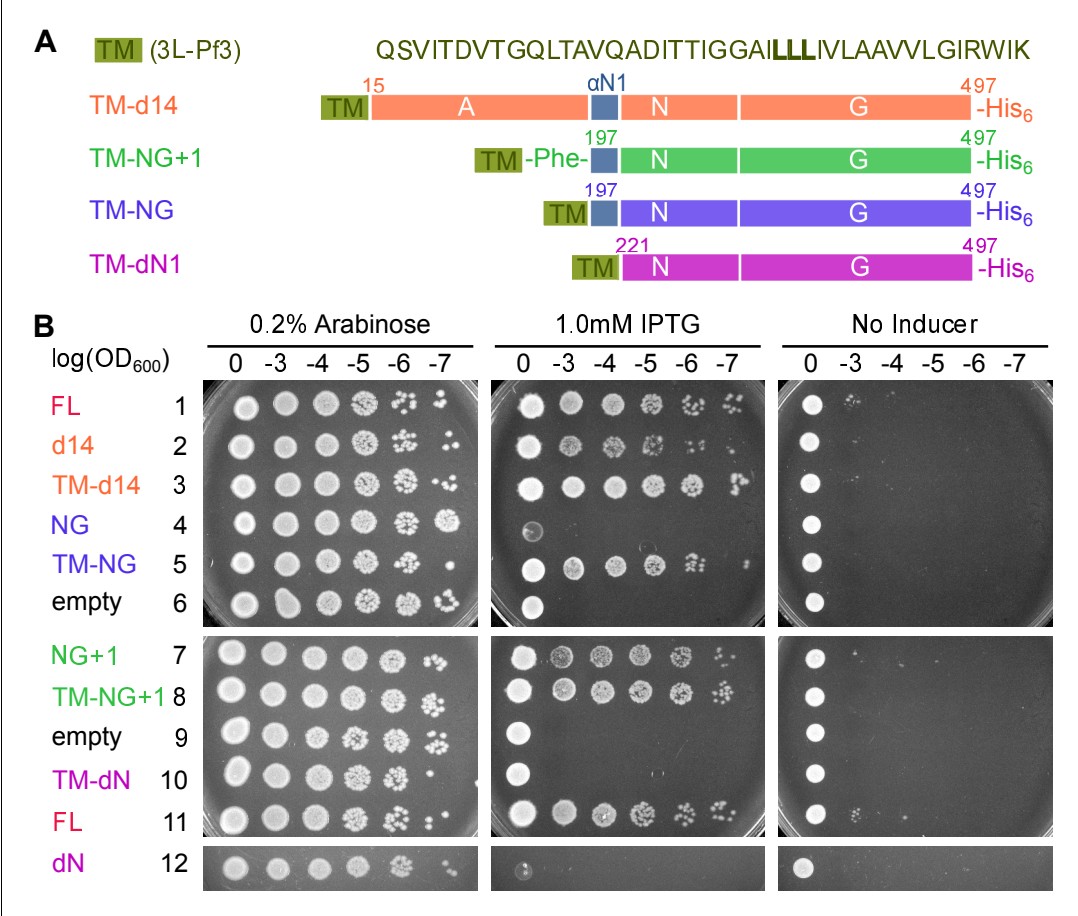

**Figure 8.** Pre-organization into the Stable mode disrupts FtsY function in vivo. (**A**) The sequence of the 3L-Pf3 TMD anchor and domain structures of membrane-tethered FtsY mutants (TM-FtsY) used in the in vivo assay. (**B**) Frogging assays were carried out to test the ability of mutant FtsYs to complement the loss of genomic FtsY, as described in Methods. Replicates of the data are shown in Figure supplement.

The following figure supplement is available for figure 8:

**Figure supplement 1.** Cell fractionation analyses and replicates of the cell growth assay.

dilution of the culture. As controls, we also tested complementation by FtsY-d14, FtsY-NG and FtsY-NG+1 with or without N-terminal fusion to 3L-Pf3 (*Figure 8*). FtsY-NG+1 showed no significant defect in supporting cell growth, as described previously (*Eitan and Bibi, 2004*; *Bahari et al., 2007*; *Parlitz et al., 2007*; *Mircheva et al., 2009*), whereas FtsY-d14 and FtsY-NG exhibited modest and strong defects, respectively (*Figure 8* and *Figure 8—figure supplement 1*). These results are consistent with the membrane binding and protein targeting activities of the respective mutants observed in vitro. Fusion to 3L-Pf3 rescued cell growth of both mutants FtsY-d14 and FtsY-NG (*Figure 8* and *Figure 8—figure supplement 1*), consistent with the robust RNC targeting activity of the corresponding tethered mutants observed in the SLB setup (*Figure 7*). This and cell fractionation analyses (*Figure 8—figure supplement 1*) corroborated that the 3L-Pf3 sequence successfully restored the membrane localization of the mutant FtsYs. In contrast, FtsY-dN1 exhibited a strong defect in supporting cell growth that was not rescued by fusion to 3L-Pf3; this defect cannot be attributed to defects in the expression or localization of TM-dN1 (*Figure 8* and *Figure 8—figure supplement 1*). The fact that TM-NG was fully functional in supporting cell growth also ruled out possible folding defects of the NG-domain due to the 3L-Pf3 fusion. Most importantly, membrane-anchored FtsY-dN1 is highly efficient in targeting correct SRP substrates (*Figure 7B and C*), indicating that deficient targeting was not responsible for the failure of TM-FtsYdN1 to support cell growth. Collectively, the

combination of in vitro and in vivo data strongly suggest that pre-organizing FtsY into the Stable mode leads to promiscuous targeting that is detrimental to cells.

On the other hand, an unfavorable pre-equilibrium to reach the stable mode in free FtsY could compromise the efficiency of the pathway if FtsY only binds membrane via the Stable mode (*Figure 1*, upper path and *Figure 9A*). We assessed whether this presents a problem for protein targeting based on the rate constants in this targeting route. The rate constant for the *early-to-closed* (or Dynamic-to-Stable) rearrangement in solution (*Figure 9A*, $k_{switch}^{cyto}$) was determined to be 0.3–0.6 s$^{-1}$ in previous studies and this work (*Figure 9C*), using acrylodan labeled at FtsY(C235) or FtsY(C356) (*Zhang et al., 2009*, *Zhang et al., 2010*; *Akopian et al., 2013a*; *Ariosa et al., 2013*). These probes specifically change fluorescence upon rearrangement of the *early* SRP•FtsY intermediate to the *closed/activated* state (*Zhang et al., 2009*; *Lam et al., 2010*). The kinetics for interaction of the *closed* SRP•FtsY complex with the membrane via the Stable mode was determined above (*Figure 3E*, $k_2$ and $k_{-2}$). Kinetic simulations using these parameters showed that targeting via this route is <50% complete in 5 s (*Figure 9D*, magenta curve), whereas the entire SRP pathway must finish within 3–5 s in bacteria before the nascent chain reaches a critical length of ~130–140 amino acids (*Noriega et al., 2014*).

We asked whether, given this unfavorable conformational pre-equilibrium in free FtsY, the Dynamic mode enables a faster targeting route compared to a route that relies solely on the Stable

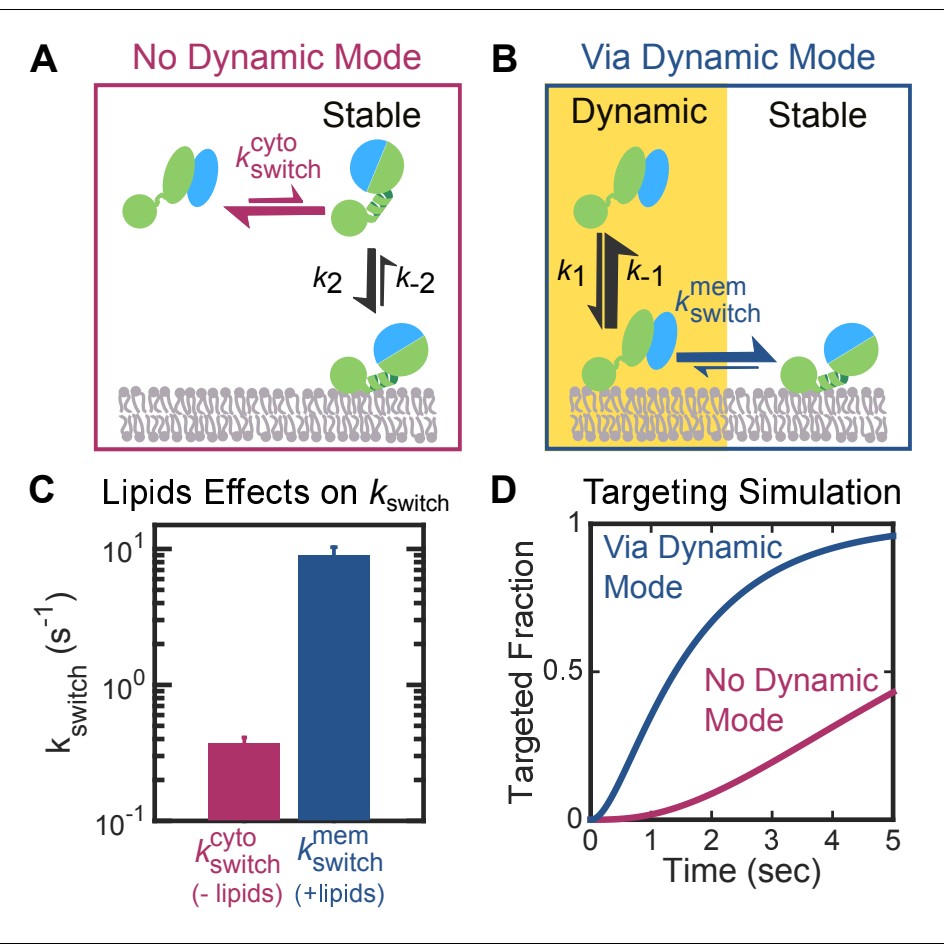

**Figure 9.** The Dynamic mode enables a faster targeting route for FtsY molecules not pre-activated at the membrane. (A, B) Depiction of the two thermodynamically equivalent routes to attain the targeting complex bound at membrane in the Stable mode. (C) Rate constants for the *early-to-closed* rearrangement in the cytosol ($k_{switch}^{cyto}$) and on the membrane ($k_{switch}^{mem}$). Values were reported as mean ± S.D. with n = 3. (D) Simulation of the kinetics of RNC targeting to membrane via the routes depicted in parts A (*magenta*) and B (*blue*).

mode (*Figure 9B*). This is probable, as in the route utilizing the Dynamic mode, the *early* complex can first associate with the membrane 10-fold more quickly (*Figure 3E*, $k_1 > k_2$). Using acrylodan-labeled FtsY(C356), we further found that lipids accelerated the rate constant of the *early*-to-*closed* rearrangement 25-fold (*Figure 9C*, $k_{switch}^{mem} > k_{switch}^{cyto}$). Kinetic simulations based on these rate constants demonstrated that the route using the Dynamic mode is five-fold faster than the alternative route (*Figure 9D*, blue curve). Thus, the Dynamic mode provides a kinetic advantage over alternative pathways that rely exclusively on the Stable mode, and alleviates the compromise in efficiency for FtsY molecules that are not pre-activated at the membrane.

## Discussion

In this work, quantitative analysis at single-molecule resolution revealed two distinct modes of membrane interactions of the bacterial SRP receptor FtsY. The Dynamic mode, characterized by rapid association with and dissociation from membrane (*Figures 1–3*), is primarily mediated by the αA1 motif (*Figure 5*). The Dynamic mode is required for FtsY to further engage with the membrane via the Stable mode, which is characterized by membrane association and dissociation kinetics 10-fold and 200-fold slower, respectively, than the Dynamic mode (*Figures 1–3*) and mediated by the previously characterized αN1 helix at the A-N domain junction (*Figure 5*). Importantly, while the Dynamic mode dominates in free FtsY and the *early* SRP•FtsY complex, the targeting complex switches to the Stable mode only when it forms the *closed* complex with SRP (*Figure 4*). These observations, together with additional findings here and previously (*Parlitz et al., 2007*; *Lam et al., 2010*; *Neher et al., 2008*; *Shepotinovskaya and Freymann, 2002*), reveal extensive auto-inhibitory mechanisms for the FtsY-membrane interaction via αN1, in contrast to the currently accepted model in which targeting occurs through FtsY molecules that are pre-bound and activated at the membrane via the αN1 motif (*Figure 1*). An engineered FtsY pre-organized into the Stable mode compromises substrate selection by the targeting pathway in vitro (*Figure 7*) and disrupts FtsY function in vivo (*Figure 8*), suggesting that pre-activating FtsY at the membrane is associated with a severe penalty. We propose that the Dynamic mode, by providing an initial membrane attachment that is uncoupled from the conformational activation of FtsY, ensures accurate substrate selection without significantly compromising the efficiency of the pathway.

Most previous work has focused on the αN1 helix of FtsY that mediates the Stable mode (*Parlitz et al., 2007*; *Lam et al., 2010*; *Stjepanovic et al., 2011*). Importantly, membrane interaction via αN1 also activates FtsY for interaction with SRP, rapid GTP hydrolysis, and subsequent cargo unloading onto the SecYEG complex (*Shepotinovskaya and Freymann, 2002*; *Gawronski-Salerno and Freymann, 2007*; *Neher et al., 2008*; *Lam et al., 2010*; *Braig et al., 2011*; *Stjepanovic et al., 2011*). This raises issues for the fidelity of substrate selection by SRP, as the kinetics of SRP-FtsY assembly is strongly regulated by the cargo and comprises a major substrate selection mechanism in the pathway (*Figure 1*, red arrows, [*Zhang et al., 2010*; *von Loeffelholz et al., 2013*]). In the extreme scenario where targeting occurs solely through FtsY molecules pre-activated at the membrane in the Stable mode prior to SRP binding, this important fidelity checkpoints would be bypassed (*Figure 1*, lower pathway). In support of this hypothesis, we found here that an engineered FtsY pre-organized into the Stable mode leads to indiscriminate targeting of SRP•FtsY complexes in vitro and cannot support cell growth in vivo. Thus, the fidelity of the SRP pathway demands that FtsY is not predominantly in the Stable/*closed* conformation before it encounters cargo-loaded SRP. Consistent with these notions, the results here and from previous work (*Lam et al., 2010*; *Stjepanovic et al., 2011*; *Neher et al., 2008*; *Draycheva et al., 2016*) show that free FtsY is extensively auto-inhibited for lipid interaction via αN1, and assembly of a *closed* complex with SRP is required to drive FtsY molecules into the Stable mode. This auto-inhibition arises not only from the acidic A-domain as shown here (*Figures 5* and *6*) but also from the N-domain of FtsY, which forms tight intra-molecular interactions with αN1 in free FtsY that reduces its accessibility (*Neher et al., 2008*; *Draycheva et al., 2016*).

Auto-inhibitory mechanisms are often associated with a penalty in efficiency. The results here show that the Dynamic mode helps alleviate this penalty, by providing a targeting route that accelerates the formation of a stably membrane-bound targeting complex compared to routes that rely only on the Stable mode. This acceleration stems from two effects: (i) membrane interaction via the Dynamic mode is ~10 fold faster than the Stable mode (*Figure 3*); and (ii) once associated with the

membrane in the Dynamic mode, the SRP•FtsY complex rearranges 25-fold more quickly into the Stable mode (*Figure 9*). More importantly, in contrast to the Stable mode, the Dynamic mode provides an initial membrane attachment without conformational activation of the receptor, and thus preserves important substrate selection mechanisms in this pathway.

We propose a revised model in which the two-step membrane binding mechanism of FtsY balances the trade-off between efficiency and selectivity (*Figure 10*). In this model, SRP initiates interaction with FtsY either in solution or bound to membrane in the Dynamic mode (*Figure 10*, events in yellow background); this generates the *early* targeting complex, which is strongly stabilized when SRP is loaded with a correct substrate (*Zhang et al., 2009*, *Zhang et al., 2010*). Once the *early* complex localizes to the membrane in the Dynamic mode, phospholipids trigger its rapid and favorable rearrangement to the *closed* state, in which FtsY further uses the αN1 motif to engage the membrane in the Stable mode, and the complex is activated for interaction with and cargo unloading onto the SecYEG translocon (*Figure 10*, events in white background). In contrast to correct substrates, the SRP•FtsY early complexes formed with incorrect substrates are much less stable and can be rejected either in solution or when bound at the membrane in the Dynamic mode (*Figure 10*, red arrows).

It is important to note that the role of FtsY's Dynamic mode in preserving substrate selection derives from its ability to provide a membrane interaction mechanism that is uncoupled from FtsY's conformational activation. Thus, the dynamic nature of this interaction is not a prerequisite for receptor molecules in general. In principle, any membrane interaction mechanism that precedes and is uncoupled from activation of downstream events could fulfill this role. This explains why replacing the αA1 motif with the 3L-Pf3 TM (this work) or with another unrelated membrane protein (*Zelazny et al., 1997*) rescued the defects of FtsY mutants lacking the Dynamic mode, as long as the mutants are not conformationally pre-organized into the *closed* state. Also consistent with this model, fusing the more hydrophobic αA1 motifs from *S. lividans* FtsY to *E. coil* FtsY-NG rescued the activity of the latter (*Bibi et al., 2001*; *Maeda et al., 2008*). It is tempting to speculate that in eukaryotic cells, although the SRP receptor is anchored at the ER membrane via the transmembrane domain of the SRβ subunit, a conceptually analogous switch in the conformation, activity, or interaction mode could be built into the eukaryotic SRP receptor (*Miller et al., 1995*).

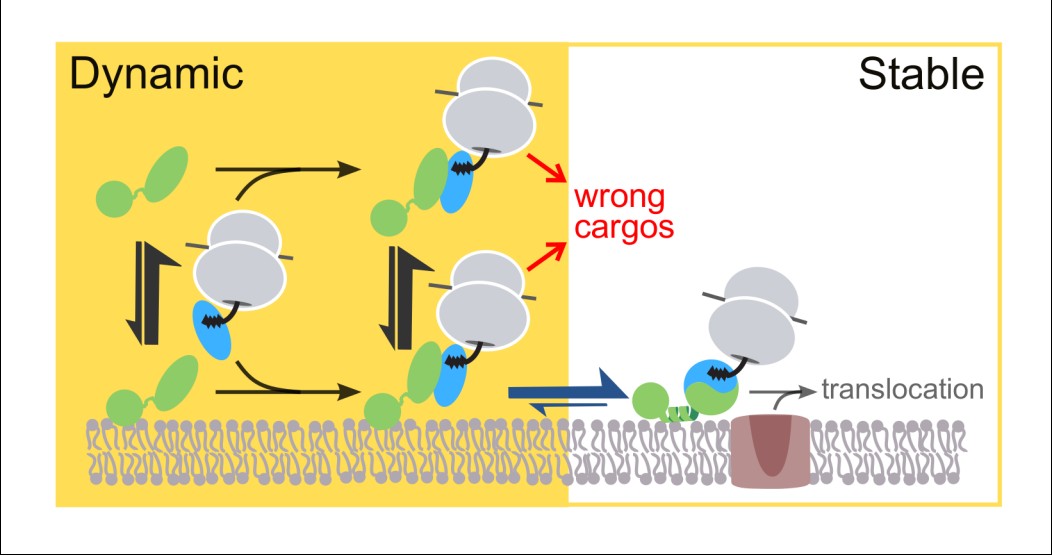

**Figure 10.** Model for the sequential membrane interaction of FtsY during protein targeting, which balances speed and specificity of the pathway. The dynamic mode mediates initial membrane association of free FtsY and the *early* SRP•FtsY complex, followed by rearrangement of the *early* complex to the *closed* state/Stable mode at the membrane. The red arrows depict rejection of SRP-independent substrates before FtsY rearranges to the *closed* state/Stable mode.

In addition to phospholipids, the SecYEG translocon has been proposed to play a crucial role in FtsY's membrane localization and in receiving cargo-loaded SRP (*Angelini et al., 2005*, *Angelini et al., 2006*; *Kuhn et al., 2015*; *Draycheva et al., 2016*). A recent study also showed that phospholipids are required for FtsY-SecYEG association (*Kuhn et al., 2015*); this places the FtsY-SecYEG interaction downstream of the FtsY-membrane interactions described here (*Figure 10*). Additionally, FtsY exhibits a strong preference for anionic phospholipids, such as PG and cardiolipin, in the Stable mode (*de Leeuw et al., 2000*; *Erez et al., 2010*; *Stjepanovic et al., 2011*). The same preference has been found for SecYEG (*Gold et al., 2010*). These observations suggest that the SecYEG translocon could further enhance the transition of FtsY to the Stable mode and vice versa, either directly through induced conformational changes or indirectly through anionic phospholipids. These remain open possibilities for future studies.

Speed and accuracy define the competency of biological systems. How to balance the trade-off between these two parameters has been widely discussed, but in few systems has this concept been studied at molecular level (*Hopfield, 1974*; *Murugan et al., 2012*). Much discussions have focused on transcription and protein synthesis machineries (*Thompson and Karim, 1982*; *Johansson et al., 2008*; *Wohlgemuth et al., 2010*; *Johansson et al., 2012*). Our work illustrates an analogous accuracy-speed tradeoff for receptor molecules. Previous models of protein targeting based solely on FtsY's Stable mode of membrane interaction exemplify extreme cases in which either speed or specificity is compromised (*Figure 1*). The two-step membrane-binding mechanism of FtsY resolves this dilemma and balances the tradeoff between efficiency and fidelity (*Figure 10*). This new model, which is largely based on energetic and kinetic principles, may provide a conceptually generalizable mechanism for membrane recruitment events in other receptor systems and targeting pathways.

## Materials and methods

### Vector, protein and RNA preparations

Plasmids for mutant FtsY were constructed using the QuikChange mutagenesis protocol (Stratagene). The expression construct for $His_6$-FtsY-dN1 was a kind gift from the Schaffitzel Lab. mRNAs for in vitro translations were synthesized by in vitro transcription using T7 (for RNC preparation) or SP6 (for co-translational translocation assay) polymerases following the Megascript protocol (Ambion). Wildtype and mutant Ffh and FtsY and 4.5S RNA were expressed and purified as described in previous studies (*Peluso et al., 2001*; *Lam et al., 2010*). RNCs bearing signal sequences of FtsQ or Luc were prepared as described previously (*Zhang et al., 2010*). FtsY-C345 and Ffh-C153 were labeled with Alexa647-maleimide (Thermo Fischer, Waltham, MA) and FtsY-C356 was labeled with acrylodan (Invitrogen) as described (*Shen et al., 2012*) with the minor modifications. Details see the SI Methods. All proteins were exchanged into SRP buffer (50 mM HEPES-KOH, pH 7.5, 150 mM KOAc, 10 mM $Mg(OAc)_2$, 2 mM dithiothreitol (DTT) and 0.01% octaethyleneglycol dodecylether (Nikkol)) prior to use.

### Fluorescence labeling

FtsY-C345 and Ffh-C153 were labeled with Alexa647-maleimide (Invitrogen) and FtsY-C356 was labeled with acrylodan (Invitrogen) as described (*Shen et al., 2012*) with the following modifications. The labeling reactions were carried out using 10-fold molar excess of Alexa647 and 30-fold molar excess of acrylodan over protein. Labeling reactions were carried out at 4°C for 2 and 16 hr for Alexa647 and acrylodan, respectively. The labeling efficiencies of each sample were quantified using the following extinction coefficients of 270,000 $cm^{-1}M^{-1}$ and 16,400 $cm^{-1}M^{-1}$ for Alexa647 and acrylodan, respectively. Protein concentrations were measured with Bradford assay using extinction coefficients of 4.8 $cm^{-1}\mu M^{-1}$ and 3.2 $cm^{-1}\mu M^{-1}$ for Ffh and FtsY, respectively.

### Supported lipid bilayer

Supported Lipid Bilayer (SLB) was prepared following established protocol with minor modifications (*Lin et al., 2010*). In brief, 1,2-dioleoyl-sn-glycero-3-phosphocholine (DOPC) and 1,2-dioleoyl-sn-glycero-3-phospho-L-serine (DOPS) chloroform stocks (Avanti Polar Lipids) were mixed in a molar ratio of 95%:5%. The lipid mixture was doped with trace amount of Texas Red 1,2-dihexadecanoyl-sn-glycero-3-phosphoethanolamine (TR-DHPE, Invitrogen) to help focus at the SLB surface. The lipid

mixture was dried at 40°C under vacuum using a rotary evaporator and stored in Argon at –30°C until use. The dried lipid film was rehydrated in ddH$_2$O to 0.5 mg/mL and sonicated at ~30% amplitude in an ice-water bath for >3 min with breaks using a microtip to generate small unilamellar vesicles (SUV).

Glass coverslips and microscopic slides were cleaned by 5 min of incubation in 3:1 vol/vol mixture of sulfuric acid/30% hydrogen peroxide, thorough rinses with ddH$_2$O, and dried under vacuum or nitrogen gas. Reaction flow chambers were assembled using the cleaned coverslips and slides. About 20 µL 0.45 mg/mL SUV suspension in TBS buffer (20 mM Tris-HCl, pH 7.5, 136 mM NaCl) was injected into the chamber and incubated at room temperature for 30–60 min. SLB was formed through self-assembly of SUVs on hydrated glass surfaces. Excess SUVs were washed out with 400 µL TBS buffer. The SLBs were imaged on the same day of preparation.

### Single-molecule instrumentation

All single-molecule assays were carried out with an objective-type total internal reflectscence microscope (Olympus X81). Green (532 nm) and red (637 nm) lasers were introduced in a 100X oil immersed objective and focused on the coverslip. Scattering light was removed by a 560 nm and a 660 nm long pass filter (Chroma) for the green and red lasers, respectively. The green laser was used to focus at SLB, which was doped with TR-DHPE. The red laser was used for imaging the protein samples. Movies were recorded using an Ixon 897 camera (Andor).

### Single-molecule imaging condition

All protein samples, except for the RNCs, were ultracentrifuged at 100,000 rpm (Optima TLX, Beckman Coulter) for 1 hr to remove aggregates. Imaging was carried out in SRP buffer supplemented with oxygen scavenging system (0.4% glucose and 1% Gloxy in Trolox [*Roy et al., 2008*]). Experiments with free FtsY-Alexa647 used imaging buffer containing 100 µM GppNHp. SRP•FtsY closed complex was assembled with 1 µM labeled FtsY-Alexa647, 3 µM Ffh, 6 µM 4.5S RNA, 100 µM GppNHp in SRP buffer and diluted to 100 pM in imaging buffer containing 100 µM GppNHp. SRP•FtsY early complex was assembled with 200 nM Ffh-Alexa647, 300 nM FtsY, 400 nM 4.5S RNA, 500 nM RNC$_{FtsQ}$ in SRP buffer without Nikkol and diluted to 100 pM in imaging buffer. The samples were then flowed onto the chamber coated with SLB for imaging. Movies were taken at a frame speed of ~20 ms/frame for 1000 frames (about 20 s) in each measurement, to minimize sample heat up and photobleaching. In each experiment, the data were averaged over movies from 10 different observation areas.

### Real-time targeting assay

Real-time targeting of SRP to the SLB was carried out using similar single-molecule imaging conditions as in the above section with the following modifications. The SLB composition for tethering FtsY-dN1 is 98% DOPC/2% Ni-NTA-DGS (1,2-dioleoyl-sn-glycero-3-[(N-(5-amino-1-carboxypentyl) iminodiacetic acid)succinyl] (nickel salt), Avanti Polar Lipids) doped with trace TR-DHPE. The Ni-SLB was first incubated with 1 µM FtsY-dN1 for 2 min in TBS buffer. Unattached FtsY-dN1 was washed out using 200 µL TBS buffer. The resulting surface density of tethered FtsY-dN1 was 3000–5000/µ m$^2$, which corresponds to a concentration in the imaging chamber of ~200 nM, for comparison with the reaction using 200 nM wildtype FtsY. Targeting reactions were initiated by mixing and injecting 200 nM wildtype FtsY and 200 pM SRP-Alexa647 loaded with RNC into SLB coated chamber, or by injecting 100 pM SRP-Alexa647 loaded with RNC into chambers in which FtsY-dN1 was tethered on the SLB. The RNC concentrations were 100 nM for FtsQ and 500 nM for Luc. The imaging chamber was connected to an automatic pump (NE-1000, New Era Pump System), which was synchronized with the camera for zero time point injection. Time-lapse images were taken at 1 s intervals with 100 ms exposure time for about 10 min. Targeting signals were quantified by counting the number of fluorescent spots on SLB from the time-lapse images. The zero drift continuous (ZDC) autofocus system was used to maintain samples in focus during injection and long-time imaging.

### Equilibrium targeting assay

The targeting selectivity of membrane-tethered FtsY-d14, FtsY-NG+1, and FtsY-NG, along with FtsY-dN1, were tested in a similar setup as in the Real-time targeting assay, where the FtsY variants

were pre-assembled to SLB, doped with 2% Ni-DGS, through their N-terminal His$_6$ tags. The tethering and targeting reaction conditions were the same as described in the previous section. As the targeting reactions finishes in 10 min (*Figure 7B*), we only recorded short movies within the 15–20 min time window of the reaction. The movies were taken at ~50 ms/frame speed for 50 frames (~2.5 secs). In each experiment, the data were averaged over movies from >6 different observation areas.

## Data processing

Trajectories of individual fluorescent spots from single-molecule experiments were extracted using a MATLAB routine combining the 'spotDetector' (*Aguet et al., 2013*) and the 'Particle Tracking' (*Blair and Dufresne, 2013*) written by Daniel Blair and Eric Dufresne. The extracted trajectories were analyzed using MATLAB. Trajectories from spots with unstable fluorescence intensity and zero mobility were discarded because they likely arise from noise and proteins aggregated at SLB defects, respectively. For real-time targeting assay, the tracking process was still carried out to identify immobile spots. Trajectories passed quality control were used for kinetic analysis as described below.

## Survival probability analysis

In most single-molecule TIRFM studies, dwell-time histograms are used for extracting kinetic parameters. This approach doesn't apply in our case since individual FtsY molecules were not tethered to the glass coverslip surface, and the short and long trajectories were unevenly sampled within the limited imaging time. Therefore, we defined a parameter, the survival probability ($P_{survival}$), for quantifying the kinetic properties of FtsY-membrane interactions on SLB.

We first define $N(i, j)$ as the number of trajectories observed at the $i^{th}$ frame and lasts another j frames. The survival probability distribution $P_{survival}(i, j)$ is then the normalized quantity:

$$P_{survival}(i, j) = \frac{N(i, j)}{N(i, 1)}. \tag{1}$$

Since the trajectories sample equilibrium distributions, which are time invariant, this function is independent of $i$ and thus simplifies to $P_{survival}(j)$, which can be obtained by time-averaging of the trajectories:

$$P_{survival}(j) = \frac{1}{m-j+1} \sum_{i=1}^{i=m-j+1} P_{survival}(i, j) \tag{2}$$

The survival probability distribution as a function of time t, $P_{survival}(t)$, was obtained by substituting the frame numbers with $t/dt$, where $dt$ is the time interval between frames. The $P_{survival}(t)$ data were fit to eq 3,

$$P_{survival}(t) = P_1 \exp(-k_{-1}t) + P_2 \exp(-k_{-2}t) \tag{3}$$

in which $k_{-1}$ and $k_{-2}$ are the dissociation rate constants of FtsY molecules from the SLB in the Dynamic and Stable modes, respectively, and P$_1$ and P$_2$ are the fraction of molecules exhibiting $k_{-1}$ and $k_{-2}$, respectively. Photobleaching was estimated from the total fluorescence intensity of fluorescently labeled FtsY, tethered on Ni-DGS SLB through the Ni-His$_6$ interaction. The timescale of photobleaching is much slower than the Stable mode and thus negligible in the analysis (*Figure 2—figure supplement 2*). However, we note that the derived value of $k_{-2}$ is close to the timescale of the slowest process that could be observed within the 20 s imaging time (to minimize sample heating and dye photobleaching). Thus, the kinetic stability of the Stable mode could be higher than the value of $k_{-2}$ reported here.

## Thermodynamic model of FtsY-Membrane interactions

To define the thermodynamic cycle of FtsY-membrane interaction, we calculated the equilibrium constants of the two membrane binding modes: $K_x = k_x/k_{-x}$, where x = 1 represents the Dynamic mode and x = 2 represents the Stable mode. The association rate constants ($k_x$) are related to the apparent association rate constants, $k_{x,app}$ for free FtsY and $k'_{x,app}$ for SRP•FtsY complex, and the fractions of the two modes in solution, f$_x$ and f'$_x$, respectively, by *Equation 4*.

$$k_{x,app} = f_x \cdot k_x$$
$$k'_{x,app} = f'_x \cdot k_x \quad x = 1,2 \tag{4}$$

Assuming FtsYs in the Dynamic and the Stable modes are the only two species in solution and their membrane-association rates and equilibrium constants are unaffected by SRP, $f_1$ and $f_2$ are constrained by mass conservation $1 = f_1 + f_2$. The same constraint is applied for $f'_1$ and $f'_2$. Substituting $f_x$ and $f'_x$ using *Equation 4* into the mass conservation equations gives *Equation 5*:

$$1 = \frac{k_{1,app}}{K_1 \cdot k_{-1}} + \frac{k_{2,app}}{K_2 \cdot k_{-2}}$$
$$1 = \frac{k'_{1,app}}{K_1 \cdot k_{-1}} + \frac{k'_{2,app}}{K_2 \cdot k_{-2}} \tag{5}$$

The values of $K_1$, $K_2$, $k_1$ and $k_2$ can be obtained by solving Eqs. 5. As the amount of membrane-bound FtsY was <1% of the total number of FtsY even in the SRP·FtsY sample, depletion of FtsY from the solution phase was therefore not considered.

The equilibrium constants between the Stable and Dynamic populations in the cytosol ($K_{cyto}$) and on the membrane ($K_{mem}$) for free FtsY are defined as:

$$K_{cyto} = \frac{f_2}{f_1}$$
$$K_{mem} = \frac{P_2}{P_1} \quad ,$$

in which $P_1$ and $P_2$ are the membrane-bound populations of FtsY in the Dynamic and Stable modes, respectively, determined from the SLB experiments using a lifetime cutoff of $\tau = 0.25$ s. The values of $f_1$ and $f_2$ were calculated using $f_1 = P_1/K_1$ and $f_2 = P_2/K_2$, respectively. The same procedure gave the values of $K'_{cyto}$, $K'_{mem}$, and $f'_x$ for the SRP·FtsY complex.

## GTPase assay

Assays to measure the stimulated GTP hydrolysis reaction between SRP and FtsY was carried out and analyzed as described (*Peluso et al., 2001*). Reaction mixtures in SRP buffer were assembled with 100 nM Ffh, 400 nM 4.5S RNA, and 0.1, 0.2, 0.5, 1, 5, 10 µM of FtsY (wildtype or mutants). Lipid-stimulated GTP hydrolysis reactions were measured using 100 nM Ffh, 400 nM 4.5S RNA, 0.2 µM FtsY (full-length or d14), and 0, 0.4, 0.6, 0.8, 1 mg/mL PG/PE liposomes. Reactions were initiated by addition of 100 µM GTP (doped with $\gamma$-$^{32}$P-GTP) and quenched with 0.75 M KH$_2$PO$_4$ (pH 3.3) at different time points. The hydrolyzed phosphate and unreacted GTP were separated by thin layer chromatography and quantified by autoradiography. The measured hydrolysis rates were fit to:

$$k_{obsd} = \frac{k_{cat}[\text{FtsY}]}{K_m + [\text{FtsY}]}, \tag{6}$$

in which $k_{cat}$ is the rate constant of GTP hydrolysis from the SRP•FtsY complex, and $k_{cat}/K_m$ approximates the association rate constant for SRP-FtsY complex formation.

## Co-translational targeting and translocation assay

Assays were carried out as described (*Shan et al., 2007*; *Shen et al., 2012*). In brief, 10 µL of in vitro translation reactions of pPL in Wheat Germ extract (Promega) containing $^{35}$S-methionine were initiated and, within 3 min of initiation, added to a mixture of 200 nM Ffh, 400 nM 4.5S RNA, 0, 14, 36, 71, 214 nM wildtype or mutant FtsY, and 0.5 eq/µL of salt-washed, trypsin-digested microsomal membrane to a total volume of <15 µL. Reactions were quenched by adding 2X SDS-loading buffer and boiling and analyzed by SDS-PAGE followed by autoradiography. The data were fit to:

$$\%translocation = \frac{V_{max}[\text{FtsY}]}{K_{1/2} + [\text{FtsY}]}, \tag{7}$$

in which $V_{max}$ is the maximum translocation efficiency at saturating concentrations of FtsY, and $K_{1/2}$ is the concentration of FtsY required to reach half of $V_{max}$.

## Liposome preparation for ensemble assays

A 70 mol% 1-palmitoyl-2-oleoyl-*sn*-glycero-3-phospho-(1'-*rac*-glycerol) (POPG) and 30 mol% 1-palmitoyl-2-oleoyl-*sn*-glycero-3-phosphoethanolamine (POPE) lipid mixture in chloroform was dried as described in SLB preparation. Dried lipid film was rehydrated in buffer containing 10 mM Tris-HCl, pH 8, and 1 mM DTT to 10 mg/mL. Large unilamellar vesicles (LUV) were generated using three-rounds of freeze-thaw cycles followed by 21 times extruding through 100 nm pore polycarbonate filters. Aliquots were flash frozen and stored at –80°C.

## Ensemble fluorescence measurements

Time-courses of early-to-closed complex rearrangement were measured on a Kintek stop-flow apparatus. The emission signal changes were monitored at 515 nm and 470 nm for samples in the absence and presence of PG/PE liposomes, respectively. The early complex was pre-assembled with 200 nM FtsY-acrylodan, 15 μM Ffh, 30 μM 4.5S RNA with or without 1 mg/mL PG/PE liposome present. The SRP concentration was varied from 15 to 25 μM to ensure complete formation of the early complex. The closed complex rearrangement was initiated by adding 200 μM GppNHp to early complex mixtures. Time courses of fluorescence changes were fit to single-exponential functions to extract the rearrangement rate constants.

## In vivo assays

Wildtype or mutant FtsY with C-terminal $His_6$-tags were cloned into pTlac18 plasmid using Gibson assembly (*Gibson et al., 2009*). For details of the construction of pTlac18 vector, see the following sections. The 3L-Pf3 DNA sequence was synthesized by standard polymerase chain reaction using overlapping oligos as described in (*Lim et al., 2013*), and cloned into FtsY constructs to make TM-fusion FtsYs. The FtsY conditional knockout strain, *E. coli* strain IY28 (*Bahari et al., 2007*), was a kind gift from the Bibi Lab. IY28 transformed with empty vector or with pTlac18 plasmids encoding wildtype or mutant FtsY were grown to log phase in 2.5 mL LB containing 0.2% arabinose, 100 μg/mL Ampicillin, 50 μg/mL Kanamycin at 37°C. The cells were harvested by low-speed centrifugation, washed once in LB, and resuspended to $OD_{600}$ = 1 in LB containing antibiotics. Serial dilutions of cell suspensions were plated in 3 μL droplets onto LB plates containing antibiotics and 0.2% arabinose or 1 mM IPTG, or no inducers. The plates were incubated at 37°C for 14 hr before imaging. Cell fractionation assays were carried out to confirm the expression and localization of the FtsY variants.

**pTlac18 Plasmid.** The pTlac18 vector is derived from pTrc99A with two modifications: (i) to reduce leaky expression from the *trc* promoter, the −35 elements were mutated from *trp* to *lacUV5* consensus sequences and the spacing between −35 and −10 elements were increased from 17 bp to 18 bp. The resulting promoter sequence is $^{-35}\underline{TT\textit{T}ACA}ATTAATCAT\textit{T}CCGGCTCG\underline{TATAAT}^{-10}$ (−35 and −10 elements are underlined and bold italic fonts indicate the mutation sites); (ii) to make a more stringent selection, an additional Kanamycin resistance site was inserted after the Ampicillin resistance site using Gibson assembly.

## Cell fractionation

Cells were inoculated into 10 mL LB containing antibiotics and 0.5 mM IPTG by diluting 1000-fold from the $OD_{600}$ = 1 suspensions. Cells were grown at 37°C to $OD_{600}$ ~1, washed twice with 10 mL LB, and then pelleted in 1.5 mL eppendorf tubes at amounts equivalent to 1 mL X 3 $OD_{600}$. The pellets were re-suspended in 900 μL lysis buffer (50 mM HEPES-KOH, pH 8, 100 mM KOAc, 10% glycerol, 1 mM DTT, 1 mM Phenylmethylsulfonyl fluoride, and protease inhibitor cocktail), incubated with 1 mg/mL lysozyme at room temperature for 30 min, and digested with DNAaseI (50 μg/mL in 16 mM $MgCl_2$) on ice for 10 min. Lysed cells were sonicated in a room temperature bath sonicator. Cell debris and unbroken cells were removed by centrifugation at 2 k rpm, 5 min in Microfuge 18 (Beckman Coulter). The total lysate sample (T) was taken from the supernatant. The inclusion body (I) was isolated by additional centrifugation at 4 k rpm for 5 min. The soluble (S) and membrane (M) fractions were further separated by ultracentrifugation at 48 k rpm for 1 hr in a TLA120.2 rotor (Optima TLX, Beckman Coulter). The inclusion body and membrane samples were dissolved in 5% SDS buffer. All fractions were analyzed using SDS-PAGE and western blotting against the $His_6$-tag.

## Kinetic simulation

Simulations were carried out in MATLAB by solving the differential equation $\dot{P}(t) = \mathbf{R}P(t)$. $P(t)$ is a vector of populations in each state (*early* and *closed* complex in cytosol or on membrane, plus a downstream targeted state) and $\mathbf{R}$ is the transition matrix composed of $k^*_1$, $k_{-1}$, $k^*_2$, $k_{-2}$, $k^{cyto}_{switch}$ and $k^{mem}_{switch}$. $k^*_1 = 1.2$ and $k^*_2 = 0.115$ (µm s$^{-1}$) are apparent association rate constants derived from $k_1$ and $k_2$, respectively, at a membrane surface area of 6 µm$^2$ and FtsY concentration of 1 µM. All these rate constants were empirically determined from the data in *Figures 3E* and *9C*. The final targeted state was simulated using a downstream reaction with rate constant of 0.7 s$^{-1}$ (*Zhang et al., 2009*), in order to drive the directionality of targeting reaction.

## Acknowledgements

We thank Connie Wang and members of the Shan lab for critical discussions and comments on the manuscript; Wen-Chen Lin, and Meredith Triplet from the Groves lab for the helps on SLB techniques; Oliver Loson from the Chan lab alumni for help on checking the SLB quality; Heun Jin Lee and Tal Einav from the Phillips Lab for helping us set up real-time delivery instruments and for modeling discussions, respectively. This work was supported by NIH grant GM078024 and the Gordon and Betty Moore Foundation through Grant GBMF2939 to SS.

## Additional information

### Funding

| Funder | Grant reference number | Author |
|---|---|---|
| National Institute of General Medical Sciences | GM078024 | Shu-ou Shan |
| Gordon and Betty Moore Foundation | GBMF2939 | Shu-ou Shan |

The funders had no role in study design, data collection and interpretation, or the decision to submit the work for publication.

### Author contributions

Y-HHF, Conceptualization, Data curation, Software, Formal analysis, Investigation, Visualization, Methodology, Writing—original draft, Writing—review and editing; WYCH, Validation, Methodology, Writing—review and editing; KS, Validation, Investigation, Methodology, Writing—review and editing; JTG, Resources, Supervision, Validation, Methodology, Writing—review and editing; TM, Formal analysis, Methodology, Writing—review and editing; S-oS, Conceptualization, Formal analysis, Supervision, Funding acquisition, Visualization, Writing—review and editing

### Author ORCIDs

Yu-Hsien Hwang Fu, http://orcid.org/0000-0002-2861-4843

Shu-ou Shan, http://orcid.org/0000-0002-6526-1733

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
