## [Decision Letter]

Thank you for submitting your article "A Two-step Membrane Binding Mechanism of the Bacterial SRP Receptor Enables Efficient and Accurate Protein Targeting" for consideration by *eLife*. Your article has been reviewed by 3 peer reviewers, one of whom is a member of our Board of Reviewing Editors, and the evaluation has been overseen Philip Cole as the Senior Editor. The reviewers have opted to remain anonymous.

The reviewers have discussed the reviews with one another and the Reviewing Editor has drafted this decision to help you prepare a revised submission.

This manuscript presents a quantitative analysis of the FtsY-lipid analysis using single-molecule spectroscopy, lipid binding, GTP hydrolysis and protein translocation assays. The authors propose that the lipid analysis occurs through a two-step mechanism in which FtsY initially interacts with the membrane dynamically (dynamic mode) whereupon the interaction of FtsY with SRP results in a more stable membrane interaction (stable mode). The latter likely primes FtsY for further downstream steps. FtsY seems to be prevented from stable binding by an auto-inhibitory mechanism, which is also required to discriminate cargo-loaded SRP. The work provides a deeper understanding of the role of the lipid interaction in SRP mediated protein targeting and provides clues on the possible mechanism of cargo rejection. However, the reviewers raise major concerns about the physiological relevance of the lipid compositions used in the study (see below).

Essential revisions:

1) It is unclear why the SLBs were prepared from DOPC/DOPS. This deviates from the native lipid composition of *E. coli*, which is PE/PG/Cardiolipin. Making SLBs from PE/PG is likely a challenge but it is essential for the authors to confirm their results with physiologically relevant lipid compositions (70/30 PE/PG lipids). As documented in a number of publications over the past two decades, the specific lipid composition is a critical factor that impacts FtsY binding and activity. The two-step binding mode observed here might be strongly influenced by the chosen lipids so that unnatural lipid composition could lead to irrelevant kinetic observations. While the authors attribute the stable mode to a conformational change induced by SRP, it could also be due to a conformational change in FtsY upon lipid interaction independent of SRP – and not physiologically significant.

Moreover, in the GTPase and fluorescence assays, "normal" composition PE/PG liposomes are used whereas for the translocation assays, microsomal membranes are employed. Therefore, these distinct experiments use different lipid compositions. As the authors state that they for the first time have conducted a rigorous and quantitative analysis of the FtsY-membrane interaction, they should be consistent in their experimental conditions. It should be kept in mind that with DOPC or POPC, the FtsY GTPase is not stimulated.

Finally, NG+1 is reported to interact weakly with the membrane itself and requires SRP to drive a favorable interaction. Again, the lipid composition used in the current study is not relevant to the in vivo situation and has been shown previously to be incompatible with stable recruitment of FtsY to the membrane.

For *E. coli*, we have a rather profound understanding of the SRP/FtsY dynamics and of the influence of lipid binding. The membrane interaction of FtsY and its consequences for the FtsY GTPase, on SRP interaction, and on protein translocation have been described previously. A quantitative analysis that introduces two steps with different affinities and kinetics would be interesting – however, only if relevant and consistent experimental conditions are used.

2) When determining lifetimes of binding to the membrane, how do the authors control for photobleaching? They should provide data to confirm that the bleaching timescale is much longer than the lifetime of the slow population.

---

## [Author Response]

Essential revisions:

1) It is unclear why the SLBs were prepared from DOPC/DOPS. This deviates from the native lipid composition of E. coli, which is PE/PG/Cardiolipin. Making SLBs from PE/PG is likely a challenge but it is essential for the authors to confirm their results with physiologically relevant lipid compositions (70/30 PE/PG lipids). As documented in a number of publications over the past two decades, the specific lipid composition is a critical factor that impacts FtsY binding and activity. The two-step binding mode observed here might be strongly influenced by the chosen lipids so that unnatural lipid composition could lead to irrelevant kinetic observations. While the authors attribute the stable mode to a conformational change induced by SRP, it could also be due to a conformational change in FtsY upon lipid interaction independent of SRP – and not physiologically significant.

*Moreover, in the GTPase and fluorescence assays, "normal" composition PE/PG liposomes are used whereas for the translocation assays, microsomal membranes are employed. Therefore, these distinct experiments use different lipid compositions. As the authors state that they for the first time have conducted a rigorous and quantitative analysis of the FtsY-membrane interaction, they should be consistent in their experimental conditions. It should be kept in mind that with DOPC or POPC, the FtsY GTPase is not stimulated.*

Finally, NG+1 is reported to interact weakly with the membrane itself and requires SRP to drive a favorable interaction. Again, the lipid composition used in the current study is not relevant to the in vivo situation and has been shown previously to be incompatible with stable recruitment of FtsY to the membrane.

The answer to these questions is that although FtsY has a preference for binding anionic phospholipids (de Leeuw et al., 2000; Erez et al., 2010; Lam et al., 2010; Stjepanovic et al., 2011), modest variations in lipid composition are insufficient to alter its mechanism of regulation shown in this work. To quickly summarize our model: (i) FtsY by itself predominantly interacts with lipids via the aA1 motif, which does not require a major conformational change in this receptor (Figure 2–Figure 5); (ii) Stable complex formation with SRP is required to induce FtsY to a distinct conformation, in which its αN1 helix is removed from inhibitory contacts with the rest of the receptor and can engage stably with membrane (Figure 2–Figure 5); (iii) Bypassing the above regulation and pre-organizing FtsY into the activated conformation compromises the fidelity of substrate selection in this pathway (Figure 7–Figure 8). Multiple observations strongly suggest that this model is robust to modest variations in lipid composition, and only breaks down under extremely unphysiological conditions. First, bacterial SRP and FtsY can replace their mammalian homologues and mediate efficient targeting and insertion of mammalian substrates into mammalian ER microsomes (Powers and Walter 1997). The substrate selection pattern is also similar between the mammalian and bacterial SRP and SRP receptors (D. Zhang and Shan 2012). This evolutionary conservation indicates that the core regulatory mechanisms of SRP and the SRP receptor are insensitive to the difference in lipid composition between the bacterial plasma membrane versus the mammalian ER membrane. Second, the GTPase and fluorescence assays demonstrating lipid-stimulation of FtsY’s biochemical activities require 70-100% PG, whereas liposomes generated from total *E. coli* lipids or 70/30 PE/PG did not give any stimulation (de Leeuw et al., 2000; Lam et al., 2010; Stjepanovic et al., 2011). These earlier results indicate that super-physiologically high PG concentrations are required for a substantial fraction of free FtsY molecules to assume the activated conformation, but most of the receptors are in the auto-inhibited state in a native *E. coli* lipid environment, as is depicted in our current model. Finally, we note that in this work, predictions generated from SLB measurements with DOPC/DOPS-lipids were tested by in vitrotargeting assays into ER microsomes and in vivocomplementation assays in bacteria, and the good agreement between the results of all three assays further supports the notion that the precise lipid composition is not a critical factor in governing the mechanism and regulation of SRP and SRP receptor. These points are now clarified under Results section.

Pragmatically, we used DOPC/DOPS in sm-TIRF measurements because this is the most robust composition to generate high quality SLBs, which is essential for quantitative single-molecule analyses (Lin et al., PNAS 2014, Huang et al., PNAS 2016, Ziemba et al., Biochemistry 2014). In contrast, PE has a much higher melting temperature than PC and is notorious for being unable to form fluid lipid bilayers. PE/PG or *E. coli* SLBs are inhomogeneous and partially in a gel-phase at room temperature (Doménech et al., 2006; Seeger et al. 2009). Some literatures even concluded that it’s impossible to form stable planar SLBs with these compositions (Dodd et al., 2008; Merz et al., 2008). Reports of complete SLB formation with *E. coli* lipids or mimics are scarce (Nollert, Kiefer, and Jähnig 1995; Garcia-Manyes, Oncins, and Sanz 2005; Lind et al., 2015), and no direct measurements to benchmark SLB quality were shown in these articles. We nevertheless attempted to form SLBs using POPE/POPG or *E. coli* lipids and found that the resulting SLBs were not mobile and full of defects, as reported.

To corroborate that our observation of SRP-induced change in FtsY-lipid interaction is robust to the difference between PG and PS (anionic lipids in bacteria and the ER membrane, respectively), we repeated the SLB measurements using DOPC/DOPG side-by-side with DOPC/DOPS. As shown in the new Figure 5—figure supplement 4, no significant difference in the distribution of FtsY in the two binding modes was observed between SLBs generated from these different lipid compositions, and this is true for both FtsY and FtsY(NG+1). The SRP-induced enrichment of the stable mode was also similar between the two different lipid compositions. Together, all the previous and current experimental data point to the fact that the SRP-mediated regulation of FtsY’s membrane interactions is fairly robust to variations in lipid composition.

*2) When determining lifetimes of binding to the membrane, how do the authors control for photobleaching? They should provide data to confirm that the bleaching timescale is much longer than the lifetime of the slow population.*

We added a bleaching curve in Figure 2—figure supplement 2. The time-scale of photobleaching is much longer than the lifetime of the stable interaction of FtsY, and is therefore negligible. This is clarified in the Survival Probability Analysis section under Material and methods.